# Neural Lyapunov Control for Discrete-Time Systems

**Junlin Wu**
Computer Science & Engineering
Washington University in St. Louis
St. Louis, MO 63130
junlin.wu@wustl.edu

**Andrew Clark**
Electrical & Systems Engineering
Washington University in St. Louis
St. Louis, MO 63130
andrewclark@wustl.edu

**Yiannis Kantaros**
Electrical & Systems Engineering
Washington University in St. Louis
St. Louis, MO 63130
ioannisk@wustl.edu

**Yevgeniy Vorobeychik**
Computer Science & Engineering
Washington University in St. Louis
St. Louis, MO 63130
yvorobeychik@wustl.edu

## Abstract

While ensuring stability for linear systems is well understood, it remains a major challenge for nonlinear systems. A general approach in such cases is to compute a combination of a Lyapunov function and an associated control policy. However, finding Lyapunov functions for general nonlinear systems is a challenging task. To address this challenge, several methods have been proposed that represent Lyapunov functions using neural networks. However, such approaches either focus on continuous-time systems, or highly restricted classes of nonlinear dynamics. We propose the first approach for learning neural Lyapunov control in a broad class of discrete-time systems. Three key ingredients enable us to effectively learn provably stable control policies. The first is a novel mixed-integer linear programming approach for verifying the discrete-time Lyapunov stability conditions, leveraging the particular structure of these conditions. The second is a novel approach for computing verified sublevel sets. The third is a heuristic gradient-based method for quickly finding counterexamples to significantly speed up Lyapunov function learning. Our experiments on four standard benchmarks demonstrate that our approach significantly outperforms state-of-the-art baselines. For example, on the path tracking benchmark, we outperform recent neural Lyapunov control baselines by an order of magnitude in both running time and the size of the region of attraction, and on two of the four benchmarks (cartpole and PVTOL), ours is the first automated approach to return a provably stable controller. Our code is available at: https://github.com/jlwu002/nlc_discrete.

## 1 Introduction

Stability analysis for dynamical systems aims to show that the system state will return to an equilibrium under small perturbations. Designing stable control in nonlinear systems commonly relies on constructing Lyapunov functions that can certify the stability of equilibrium points and estimate their region of attraction. However, finding Lyapunov functions for arbitrary nonlinear systems is a challenging task that requires substantial expertise and manual effort [Khalil, 2015, Lavaei and Bridgeman, 2023]. To address this challenge, recent progress has been made in learning Lyapunov functions in *continuous-time* nonlinear dynamics [Abate et al., 2020, Chang et al., 2019, Zhou et al., 2022]. However, few approaches exist for doing so in discrete-time systems [Dai et al., 2021], and none for general Lipschitz-continuous dynamics. Since modern learning-based controllers often

37th Conference on Neural Information Processing Systems (NeurIPS 2023).

take non-negligible time for computation, the granularity of control is effectively discrete-time, and developing approaches for stabilizing such controllers is a major open challenge.

We propose a novel method to learn Lyapunov functions and stabilizing controllers, represented as neural networks (NNs) with ReLU activation functions, for discrete-time nonlinear systems. The proposed framework broadly consists of a *learner*, which uses a gradient-based method for updating the parameters of the Lyapunov function and policy, and a *verifier*, which produces counterexamples (if any exist) to the stability conditions that are added as training data for the learner. The use of a verifier in the learning loop is critical to enable the proposed approach to return a provably stable policy. However, no prior approaches enable sound verification of neural Lyapunov stability for general discrete-time dynamical systems. The closest is Dai et al. [2021], who assume that dynamics are represented by a neural network, an assumption that rarely holds in real systems. On the other hand, approaches for continuous-time systems [Chang et al., 2019, Zhou et al., 2022] have limited efficacy in discrete-time environments, as our experiments demonstrate. To address this gap, we develop a novel verification tool that checks if the candidate NN Lyapunov function satisfies the Lyapunov conditions by solving Mixed Interger Linear Programs (MILPs). This approach, which takes advantage of the structure of discrete-time Lyapunov stability conditions, can soundly verify a broad class of dynamical system models.

However, using a sound verification tool in the learning loop makes it a significant bottleneck, severely limiting scalability. We address this problem by also developing a highly effective gradient-based technique for identifying counterexamples, resorting to the full MILP-based verifier only as a last resort. The full learning process thereby iterates between learning and verification steps, and returns only when the sound verifier is able to prove that the Lyapunov function and policy satisfy the stability conditions.

The final technical challenge stems from the difficulty of verifying stability near the origin [Chang et al., 2019], typically addressed heuristically by either adding a fixed tolerance to a stability condition [Dai et al., 2021], or excluding a small area around the origin from verification [Chang et al., 2019, Zhou et al., 2022]. We address it by adapting Lyapunov stability theory to ensure convergence to a small region near the origin, thereby achieving the first (to our knowledge) sound approach for computing a stable neural controller that explicitly accounts for such approximations near the origin.

We evaluate the proposed approach in comparison to state-of-the-art baselines on four standard nonlinear control benchmarks. On the two simpler domains (inverted pendulum and path following), our approach outperforms the state-of-the-art continuous-time neural Lyapunov control approaches by at least several factors and up to an order of magnitude *in both running time and the size of the region of attraction*. On the two more complex domains—cartpole and PVTOL—ours *is the first automated approach that returns a provably stable controller*. Moreover, our ablation experiments demonstrate that both the MILP-based verifier and heuristic counterexample generation technique we propose are critical to the success of our approach.

In summary, we make the following contributions:

- A novel MILP-based approach for verifying a broad class of discrete-time controlled nonlinear systems.
- A novel approach for learning provably verified stabilizing controllers for a broad class of discrete-time nonlinear systems which combines our MILP-based verifier with a heuristic gradient-based approximate counterexample generation technique.
- A novel formalization of approximate stability in which the controller provably converges to a small ball near the origin in finite time.
- Extensive experiments on four standard benchmarks demonstrate that by leveraging the special structure of Lyapunov stability conditions for discrete-time system, our approach significantly outperforms prior art.

**Related Work**  Much prior work on learning Lyapunov functions focused on continuous-time systems [Abate et al., 2020, Ravanbakhsh and Sankaranarayanan, 2019, Chang et al., 2019, Zhou et al., 2022]. Common approaches have assumed that dynamics are linear [Donti et al., 2021, Tedrake, 2009] or polynomial [Ravanbakhsh and Sankaranarayanan, 2019]. Recently, Chang et al. [2019], Rego and de Araújo [2022], and Zhou et al. [2022] proposed learning Lyapunov functions represented as neural networks while restricting policies to be linear. These were designed for continuous-time dynamics, and are not effective in discrete-time settings, as we show in the experiments. Chen et al.

[2021a] learns convex Lyapunov functions for discrete-time hybrid systems. Their approach requires hybrid systems to admit a mixed-integer linear programming formulation, essentially restricting it to piecewise-affine systems, and does not learn stabilizing controllers for these. Our approach considers a much broader class of dynamical system models, and learns provably stable controllers and Lyapunov functions, allowing both to be represented as neural networks. Kolter and Manek [2019] learn stable nonlinear dynamics represented as neural networks, but do not provide stability guarantees with respect to the true underlying system. In addition, several approaches have been proposed that either provide only probabilistic stability guarantees [Berkenkamp et al., 2017, Richards et al., 2018], or do not guarantee stability [Choi et al., 2020, Han et al., 2020]. Several recent approaches propose methods for verifying stability in dynamical discrete-time systems. Chen et al. [2021b] compute an approximate region of attraction of dynamical systems with neural network dynamics, but assume that Lyapunov functions are quadratic. Dai et al. [2021] consider the problem of verifying Lyapunov conditions in discrete-time systems, as well as learning provably stable policies. However, they assume that dynamics are represented by neural networks with (leaky) ReLU activation functions. Our verification approach, in contrast, is for arbitrary Lipschitz continuous dynamics.

## 2 Model

We consider a discrete-time nonlinear dynamical system

$$x_{t+1} = f(x_t, u_t), \tag{1}$$

where $x_t \in \mathcal{X}$ is state in a domain $\mathcal{X}$ and $u_t$ the control input at time $t$. We assume that $f$ is Lipschitz continuous with Lipschitz constant $L_f$. This class of dynamical systems includes the vast majority of (non-hybrid) dynamical system models in prior literature. We assume that $x = 0$ is an equilibrium point for the system, that is, $f(0, u_0) = 0$ for some $u_0$. Let $\pi(x)$ denote a control policy, with $u_t = \pi(x_t)$. For example, in autonomous vehicle path tracking, $x$ can measure path tracking error and $u$ the steering angle. In the conventional setup, the goal is to learn a control policy $\pi$ such that the dynamical system in Equation (1) converges to the equilibrium $x = 0$, a condition referred to as *stability*. To this end, we can leverage the Lyapunov stability framework [Tedrake, 2009]. Specifically, the goal is to identify a Lyapunov function $V(x)$ and controller $\pi$ that satisfies the following conditions over a subset of the domain $\mathcal{R} \subseteq \mathcal{X}$: 1)$V(0) = 0$; 2)$V(x) > 0, \forall x \in \mathcal{R} \setminus \{0\}$; and 3)$V(f(x, \pi(x))) - V(x) < 0, \forall x \in \mathcal{R}$. These conditions imply that the system is (locally) stable in the Lyapunov sense [Bof et al., 2018, Tedrake, 2009].

In practice, due to numerical challenges in verifying stability conditions near the origin, it is common to verify slight relaxations of the Lyapunov conditions. These typically fall into two categories: 1) Lyapunov conditions are only checked for $\|x\|_p \geq \epsilon$ [Chang et al., 2019, Zhou et al., 2022] (our main baselines, and the only other methods for learning neural Lyapunov control for general nonlinear dynamics), and/or 2) a small tolerance $\delta > 0$ is added in verification, allowing small violations of Lyapunov conditions near the origin [Dai et al., 2021].

Our goal now is to formally investigate the implications of numerical approximations of this kind. We next define a form of stability which entails finite-time convergence to $\mathcal{B}(0, \epsilon) = \{x \mid \|x\|_\infty < \epsilon\}$.

**Definition 2.1** ($\epsilon$-stability). *We call pair of $(V, \pi)$ $\epsilon$-stable within a region $\mathcal{R}$ when the following conditions are satisfied: (a) $V(0) = 0$; (b) there exists $\zeta > 0$ such that $V(f(x, \pi(x))) - V(x) < -\zeta$ for all $x \in \mathcal{R} \setminus \mathcal{B}(0, \epsilon)$; and (c) $V(x) > 0$ for all $x \in \mathcal{R} \setminus \mathcal{B}(0, \epsilon)$.*

A key ingredient to achieving stability is to identify a *region of attraction (ROA)*, within which we are guaranteed to converge to the origin from any starting point. In the context of $\epsilon$-stability, our goal is to converge to a small ball near the origin, rather than the origin; let $\epsilon$-ROA be the set of initial inputs that has this property. In order to enable us to identify an $\epsilon$-ROA, we introduce a notion of an invariant sublevel set. Specifically, we refer to a set $\mathcal{D}(\mathcal{R}, \rho) = \{x \in \mathcal{R} | V(x) \leq \rho\}$ with the property that $x \in \mathcal{D}(\mathcal{R}, \rho) \Rightarrow f(x, \pi(x)) \in \mathcal{R}$ as a *$\mathcal{R}$-invariant sublevel set*. In other words, $\mathcal{D}(\mathcal{R}, \rho)$ is a sublevel set which is additionally forward invariant with respect to $\mathcal{R}$. We assume that $\mathcal{B}(0, \epsilon) \subset \mathcal{D}(\mathcal{R}, \rho)$.

Next, we formally prove that $\epsilon$-stability combined with a sublevel set $\mathcal{D}(\mathcal{R}, \rho)$ entails convergence in three senses: 1) that we reach an $\epsilon$-ball around the origin in finite time, 2) that we reach it infinitely often, and 3) we converge to an arbitrarily small ball around the origin. We refer to the first of these senses as *reachability*. What we show is that for $\epsilon$ sufficiently small, reachability implies convergence in all three senses. For this, a key additional assumption is that $V$ and $\pi$ are Lipschitz continuous, with Lipschitz constants $L_v$ and $L_\pi$, respectively (e.g., in ReLU neural networks).

**Theorem 2.2.** *Suppose $V$ and $\pi$ are $\epsilon$-stable on a compact $\mathcal{R}$, $\mathcal{D}(\mathcal{R}, \rho)$ is an $\mathcal{R}$-invariant set, and $\exists c_1$ such that $\|\pi(0) - u_0\|_\infty \le c_1 \epsilon$. Then if $x_0 \in \mathcal{D}(\mathcal{R}, \rho) \setminus \mathcal{B}(0, \epsilon)$:*

> *(i) there exists a finite $K$ such that $x_K \in \mathcal{B}(0, \epsilon)$,*
> *(ii) $\exists c_2$ such that if $c_2 \epsilon < \rho$ and $\mathcal{B}(0, c_2 \epsilon) \subset \mathcal{R}$, then there exists a finite $K$ such that $\forall k \ge K$, $x_k \in \mathcal{D}(\mathcal{R}, c_2 \epsilon)$ and the sequence $\{x_k\}_{k \ge 0}$ reaches $\mathcal{B}(0, \epsilon)$ infinitely often, and furthermore*
> *(iii) for any $\eta > 0$ such that $\mathcal{B}(0, \eta) \subset \mathcal{R}$, $\exists \epsilon$ and finite $K$ such that $\|\pi(0) - u_0\|_\infty \le c_1 \epsilon \Rightarrow \|x_k\|_\infty \le \eta \; \forall k \ge K$.*

*Proof.* We prove (i) by contradiction. Suppose that $\|x_k\|_\infty > \epsilon \; \forall k \in \{0, \ldots, \lceil V(x_0)/\zeta \rceil\}$. Then, when $k = \lceil V(x_0)/\zeta \rceil$, condition (b) of $\epsilon$-stability and $\mathcal{R}$-invariance of $\mathcal{D}(\mathcal{R}, \rho)$ implies that $V(x_k) < V(x_0) - \zeta k < 0$, contradicting condition (c) of $\epsilon$-stability.

To prove (ii), fix the finite $K$ from (i), and let $k \ge K$ be such that $x_k \in \mathcal{B}(0, \epsilon)$. By Lipschitz continuity of $V$ and the fact that $V(0) = 0$ and $V(x) \ge 0$ (conditions (a) and (c) of $\epsilon$-stability), $V(x_{k+1}) \le L_v \|x_{k+1}\|$, where $\| \cdot \|$ is the $\ell_\infty$ norm here and below. Moreover, since $x_{k+1} = f(x_k, \pi(x_k))$, $f(0, u_0) = 0$ (stability of the origin), and by Lipschitz continuity of $f$ and $\pi$,

$$\|x_{k+1}\| \le L_f \|(x_k, \pi(x_k) - u_0)\| \le L_f \max\{\|x_k\|, \|\pi(x_k) - u_0\|\}.$$

By Lipschitz continuity of $\pi$, triangle inequality, and the condition that $\|\pi(0) - u_0\| \le c_1 \epsilon$, we have $\|\pi(x_k) - u_0\| \le L_\pi \|x_k\| + c_1 \epsilon \le (L_\pi + c_1)\epsilon$. Let $c_2 = \max\{L_v, 1\} L_f \max\{1, L_\pi + c_1\}$. Then $\|x_{k+1}\| \le c_2 \epsilon$ and $V(x_{k+1}) \le c_2 \epsilon$. Thus, if $c_2 \epsilon < \rho$ and $\mathcal{B}(0, c_2 \epsilon) \subset \mathcal{R}$, and $\mathcal{D}(\mathcal{R}, \rho)$ is $\mathcal{R}$-invariant, then $x_{k+1} \in \mathcal{D}(\mathcal{R}, c_2 \epsilon)$ which is $\mathcal{R}$-invariant. Additionally, either $x_{k+1} \in \mathcal{B}(0, \epsilon)$, and by the argument above $x_{k+2} \in \mathcal{D}(\mathcal{R}, c_2 \epsilon)$, or $x_{k+1} \in \mathcal{D}(\mathcal{R}, c_2 \epsilon) \setminus \mathcal{B}(0, \epsilon)$, and by $\epsilon$-stability $x_{k+2} \in \mathcal{D}(\mathcal{R}, c_2 \epsilon)$. Thus, by induction, we have that for all $k \ge K$, $x_k \in \mathcal{D}(\mathcal{R}, c_2 \epsilon)$. Finally, since for all $k \ge K$, $x_k \in \mathcal{D}(\mathcal{R}, c_2 \epsilon)$, if $x_k \notin \mathcal{B}(0, \epsilon)$, by (i) it must reach $\mathcal{B}(0, \epsilon)$ in finite time. Consequently, there is an infinite subsequence $\{x_{k'}\}$ of $\{x_k\}_{k \ge 0}$ such that all $x_{k'} \in \mathcal{B}(0, \epsilon)$, that is, the sequence $\{x_k\}_{k \ge 0}$ reaches $\mathcal{B}(0, \epsilon)$ infinitely often.

We prove part (iii) by contradiction. Fix $\eta > 0$ and define $S = \{x \in \mathcal{R} : \|x\| > \eta\}$. Suppose that $\forall \epsilon > 0$ there exists $x \in S$ such that $V(x) \le c_2 \epsilon$ where $c_2$ is as in (ii). Then for any (infinite) sequence of $\{\epsilon_t\}$ we have $\{x_t\}$ such that $V(x_t) \le c_2 \epsilon_t$, where $x_t \in S$. Now, consider a set $\bar{S} = \{x \in \mathcal{R} : \|x\| \ge \eta\}$. Since $\bar{S}$ is compact and $\{x_t\}$ is an infinite sequence, there is an infinite subsequence $\{x_{t_k}\}$ of $\{x_t\}$ such that $\lim_{k \to \infty} x_{t_k} = x^*$ and $x^* \in \bar{S}$. Since $V$ is continuous, we have $\lim_{k \to \infty} V(x_{t_k}) = V(x^*)$. Now, we choose $\{\epsilon_t\}$ such that $\lim_{t \to \infty} \epsilon_t = 0$. This means that $\lim_{k \to \infty} V(x_{t_k}) = V(x^*) \le 0$, and since $x^* \in \bar{S}$, this contradicts condition (c) of $\epsilon$-stability. Since by (ii), there exists a finite $K$, $V(x_k) \le c_2 \epsilon, \forall k \ge K$, we have $\forall k \ge K$, $\|x_k\| \le \eta$. $\qquad \square$

The crucial implication of Theorem 2.2 is that as long as we choose $\epsilon > 0$ sufficiently small, verifying that $V$ and $\pi$ are $\epsilon$-stable together with identifying an $\mathcal{R}$-invariant set $\mathcal{D}(\mathcal{R}, \rho)$ suffices for convergence arbitrarily close to the origin. One caveat is that we need to ensure that $\pi(0)$ is sufficiently close to $u_0$ for any $\epsilon$ we choose. In most domains, this can be easily achieved: for example, if $u_0 = 0$ (as is the case in many settings, including three of our four experimental domains), we can use a representation for $\pi$ with no bias terms, so that $\pi(0) = 0 = u_0$ by construction. In other cases, we can simply check this condition after learning.

Another caveat is that we need to define $\mathcal{R}$ to enable us to practically achieve these properties. To this end, we define $\mathcal{R}$ parametrically as $\mathcal{R}(\gamma) = \{x \in \mathcal{X} \mid \|x\|_\infty \le \gamma\}$. Additionally, we introduce the following useful notation. Define $\mathcal{R}(\epsilon, \gamma) = \{x \in \mathcal{X} \mid \epsilon \le \|x\|_\infty \le \gamma\}$. Note that $\mathcal{R}(0, \infty) = \mathcal{X}$, $\mathcal{R}(\epsilon, \infty) = \{x \in \mathcal{X} \mid \|x\|_\infty \ge \epsilon\}$ and $\mathcal{R}(0, \gamma) = \mathcal{R}(\gamma)$. Thus, for $\gamma$ sufficiently large compared to $\epsilon$, conditions such as $\mathcal{B}(0, \epsilon) \subset \mathcal{D}(\mathcal{R}, \rho)$, $\mathcal{B}(0, \eta) \subset \mathcal{R}$, and $\mathcal{B}(0, c_2 \epsilon) \subset \mathcal{R}$ will be easy to satisfy. Following conventional naming, we denote $\mathcal{R}(\epsilon, \gamma)$ as *$\epsilon$-valid region, i.e., the region that satisfies conditions $(a) - (c)$ of $\epsilon$-stability, and refer to a function $V$ satisfying these conditions as an $\epsilon$-Lyapunov function.*

We assume that the $\epsilon$-Lyapunov function as well as the policy can be represented in a parametric function class, such as by a deep neural network. Formally, we denote the parametric $\epsilon$-Lyapunov function by $V_\theta(x)$ and the policy by $\pi_\beta(x)$, where $\theta$ and $\beta$ are their respective parameters. Let $\mathcal{D}(\gamma, \rho) \equiv \mathcal{D}(\mathcal{R}(\gamma), \rho)$. Our goal is to learn $V_\theta$ and $\pi_\beta$ such we maximize the size of the $\mathcal{R}(\gamma)$-invariant sublevel set $\mathcal{D}(\gamma, \rho)$ such that $V_\theta$ and $\pi_\beta$ are provably $\epsilon$-stable on $\mathcal{R}(\gamma)$. Armed with Theorem 2.2, we refer to this set simply as ROA below for simplicity and for consistency with prior work, e.g., [Chang et al., 2019, Zhou et al., 2022], noting all the caveats discussed above.

Define $\mathcal{P}(\gamma)$ as the set $(\theta, \beta)$ for which the conditions (a)-(c) of $\epsilon$-stability are satisfied over the domain $\mathcal{R}(\gamma)$. Our main challenge below is to find $(\theta, \beta) \in \mathcal{P}(\gamma)$ for a given $\gamma$. That, in turn, entails solving the key subproblem of verifying these conditions for given $V_\theta$ and $\pi_\beta$.

Next, in Section 3 we address the verification problem, and in Section 4 we describe our approach for jointly learning $V_\theta$ and $\pi_\beta$ that can be verified to satisfy the $\epsilon$-stability conditions for a given $\mathcal{R}(\gamma)$.

## 3   Verifying Stability Conditions

Prior work on learning $\epsilon$-Lyapunov functions for continuous-time nonlinear control problems has leveraged off-the-shelf SMT solvers, such as dReal [Gao et al., 2013]. However, these solvers scale poorly in our setting (see Supplement C for details). In this section, we propose a novel approach for verifying the $\epsilon$-Lyapunov conditions for arbitrary Lipschitz continuous dynamics using mixed-integer linear programming, through obtaining piecewise-linear bounds on the dynamics. We assume $V_\theta$ and $\pi_\beta$ are $K$- and $N$-layer neural networks, respectively, with ReLU activations.

We begin with the problem of verifying condition (c) of $\epsilon$-stability, which we represent as a feasibility problem: to find if there is any point $\tilde{x} \in \mathcal{R}(\epsilon, \gamma)$ such that $V(\tilde{x}) \leq 0$. We can formulate it as the following MILP:

$$z_K \leq 0 \tag{2a}$$
$$z_{l+1} = g_{\theta_l}(z_l), \quad 0 \leq l \leq K - 1 \tag{2b}$$
$$\epsilon \leq \|x\|_\infty \leq \gamma, \quad z_0 = x, \tag{2c}$$

where $l$ refers to a layer in the neural network $V_\theta$, $z_K = V_\theta(x)$, and the associated functions $g_{\theta_l}(z_l)$ are either $W_l z_l + b_l$ for a linear layer (with $\theta_l = (W_l, b_l)$) or $\max\{z_l, 0\}$ for a ReLU activation. Any feasible solution $x^*$ is then a counterexample, and if the problem is infeasible, the condition is satisfied. ReLU activations $g(z)$ can be linearized by introducing an integer variable $a \in \{0, 1\}$, and replacing the $z' = g(z)$ terms with constraints $z' \leq Ua$ and $z' \leq -L(1 - a)$, where $L$ and $U$ are specified so that $L \leq z \leq U$ (we deal with identifying $L$ and $U$ below).

Next, we cast verification of condition $(b)$ of $\epsilon$-stability, which involves the nonlinear control dynamics $f$, as the following feasibility problem:

$$\bar{z}_K - z_K \geq -\zeta \tag{3a}$$
$$y_{i+1} = h_{\beta_i}(y_i), \quad 0 \leq i \leq N - 1 \tag{3b}$$
$$\bar{z}_{l+1} = g_{\theta_l}(\bar{z}_l), \quad 0 \leq l \leq K - 1 \tag{3c}$$
$$\bar{z}_0 = f(x, y_N) \tag{3d}$$
$$y_0 = x, \quad \text{constraints } (2b) - (2c), \tag{3e}$$

where $h_{\beta_i}()$ are functions computed at layers $i$ of $\pi_\beta$, $z_K = V_\theta(x)$, and $\bar{z}_K = V_\theta(f(x, \pi_\beta(x)))$.

At this point, all of the constraints can be linearized as before with the exception of Constraint (3d), which involves the nonlinear dynamics $f$. To deal with this, we relax the verification problem by replacing $f$ with linear lower and upper bounds. To obtain tight bounds, we divide $\mathcal{R}(\gamma)$ into a collection of subdomains $\{\mathcal{R}_k\}$. For each $\mathcal{R}_k$, we obtain a linear lower bound $f_{lb}(x)$ and upper bound $f_{ub}(x)$ on $f$, and relax the problematic Constraint (3d) into $f_{lb}(x) \leq \bar{z}_0 \leq f_{ub}(x)$, which is now a pair of linear constraints. We can then solve Problem (3) for each $\mathcal{R}_k$.

**Computing Linear Bounds on System Dynamics** Recall that $f : \mathbb{R}^n \mapsto \mathbb{R}^n$ is the Lipschitz-continuous system dynamic $x_{t+1} = f(x_t, u_t)$. For simplicity we omit $u_t = \pi_\beta(x)$ below. Let $\lambda$ be the $\ell_\infty$ Lipschitz constant of $f$. Suppose that we are given a region of the domain $\mathcal{R}_k$ represented as a hyperrectangle, i.e., $\mathcal{R}_k = \times_i [x_{i,l}, x_{i,u}]$, where $[x_{i,l}, x_{i,u}]$ are the lower and upper bounds of $x$ along coordinate $i$. Our goal is to compute a linear upper and lower bound on $f(x)$ over this region. We bound $f_j(x) : \mathbb{R}^n \mapsto \mathbb{R}$ along each $j$-th dimension separately. By $\lambda$-Lipschitz-continuity, we can obtain $f_j(x) \leq f_j(x_{1,l}, x_{2,l}, \ldots, x_{n,l}) + \lambda \sum_i (x_i - x_{i,l})$ and $f_j(x) \geq f_j(x_{1,l}, x_{2,l}, \ldots, x_{n,l}) - \lambda \sum_i (x_i - x_{i,l})$. The full derivation is provided in the Supplement Section B.

Alternatively, if $f_j$ is differentiable, convex over $x_i$, and monotone over other dimensions, we can restrict it to calculating one-dimensional bounds by finding coordinates $x_{-i,u}$ and $x_{-i,l}$ such that $f_j(x_i, x_{-i,l}) \leq f_j(x_i, x_{-i}) \leq f_j(x_i, x_{-i,u})$ for any given $x$. Then $f_j(x) \leq f_{j,i,ub}(x) \equiv f_j(x_{i,l}, x_{-i,u}) + \frac{f_j(x_{i,u}, x_{-i,u}) - f_j(x_{i,l}, x_{-i,u})}{x_{i,u} - x_{i,l}}(x_i - x_{i,l})$ and $f_j(x) \geq f_{i,lb}(x) \equiv f_j(x_i^*, x_{-i,l}) +$

$f'_j(x^*_i, x_{-i,l})(x_i - x^*_i)$, where we let $x^*_i = \min_{x_s \in [x_{i,l}, x_{i,u}]} \int_{x_{i,l}}^{x_{i,u}} [f_j(x_i, x_{-i,l}) - f_j(x_s, x_{-i,l}) + f'_j(x_s, x_{-i,l})(x_i - x_s)] dx_s$ to minimize the error area between $f_j(x)$ and $f_{i,lb}(x)$. Note that the solution for $x^*_i$ can be approximated when conducting experiments, and this approximation has no impact on the correctness of the bound. A similar result obtains for where $f_j(x)$ is concave over $x_i$. Furthermore, we can use these single-dimensional bounds to obtain tighter lower and upper bounds on $f_j(x)$ as follows: note that for any $c^l, c^u$ with $\sum_i c^l_i = 1$ and $\sum_i c^u_i = 1$, $\sum_i c^l_i f_{j,i,lb}(x) \le f_j(x) \le \sum_i c^u_i f_{j,i,ub}(x)$, which means we can optimize $c^l$ and $c^u$ to minimize the bounding error. In practice, we can typically partition $\mathcal{R}(\gamma)$ so that the stronger monotonicity and convexity or concavity assumptions hold for each $\mathcal{R}_k$.

Note that our linear bounds $f_{lb}(x)$ and $f_{ub}(x)$ introduce errors compared to the original nonlinear dynamics $f$. However, we can obtain tighter bounds by splitting $\mathcal{R}_k$ further into subregions, and computing tighter bounds in each of the resulting subregions, but at the cost of increased computation. To balance these considerations, we start with a relatively small collection of subdomains $\{\mathcal{R}_k\}$, and only split a region $\mathcal{R}_k$ if we obtain a counterexample in $\mathcal{R}_k$ that is not an actual counterexample for the true dynamics $f$.

**Computing Bounds on ReLU Linearization Constants**  In linearizing the ReLU activations, we supposed an existence of lower and upper bounds $L$ and $U$. However, we cannot simply set them to some large negative and positive number, respectively, because $V_\theta(x)$ has no a priori fixed bounds (in particular, note that for any $\epsilon$-Lyapunov function $V$ and constant $a > 1$, $aV$ is also a $\epsilon$-Lyapunov function). Thus, arbitrarily setting $L$ and $U$ makes verification unsound. To address this issue, we use interval bound propogation (IBP) [Gowal et al., 2018] to obtain $M = \max_{1 \le i \le n} \{|U_i|, |L_i|\}$, where $U_i$ is the upper bound, and $L_i$ is the lower bound returned by IBP for the $i$-th layer, with inputs for the first layer the upper and lower bounds of $f(\mathcal{R}(\gamma))$. Setting each $L = -M$ and $U = M$ then yields sound verification.

**Computing Sublevel Sets**  The approaches above verify the conditions (a)-(c) of $\epsilon$-stability on $\mathcal{R}(\gamma)$. The final piece is to find the $\mathcal{R}(\gamma)$-invariant sublevel set $\mathcal{D}(\gamma, \rho)$, that is, to find $\rho$. Let $B(\gamma) \ge \max \left( \max_{x \in \mathcal{R}(\gamma)} \|f(x, \pi_\beta(x))\|_\infty, \gamma \right)$. We find $\rho$ by solving

$$\min_{x : \gamma \le \|x\|_\infty \le B(\gamma)} V(x). \tag{4}$$

We can transform both Problem (4) and computation of $B(\gamma)$ into MILP as for other problems above.

**Theorem 3.1.** *Suppose that $V$ and $\pi$ are $\epsilon$-stable on $\mathcal{R}(\gamma)$, $\|\pi(0) - u_0\|_\infty \le \epsilon$, and $\gamma \ge L_f \max\{1, L_\pi + 1\}\epsilon$. Let $V^*$ be the optimal value of the objective in Problem (4), and $\rho = V^* - \mu$ for any $\mu > 0$. Then the set $\mathcal{D}(\gamma, \rho) = \{x : x \in \mathcal{R}(\gamma), V(x) \le \rho\}$ is an $\mathcal{R}(\gamma)$-invariant sublevel set.*

*Proof.* If $\|x\|_\infty > \epsilon$, $V(f(x, \pi(x))) < V(x) \le \rho < V^*$ by $\epsilon$-stability of $V$ and $\pi$. Suppose that $\bar{x} = f(x, \pi(x)) \notin \mathcal{R}(\gamma)$. Since $V(\bar{x}) < V^*$, $V^*$ must not be an optimal solution to (4), a contradiction. If $\|x\|_\infty \le \epsilon$, the argument is similar to Theorem 2.2 (ii). □

## 4 Learning $\epsilon$-Lyapunov Function and Policy

We cast learning the $\epsilon$-Lyapunov function and policy as the following problem:

$$\min_{\theta, \beta} \sum_{i \in S} \mathcal{L}(x_i; V_\theta, \pi_\beta), \tag{5}$$

where $\mathcal{L}(\cdot)$ is a loss function that promotes Lyapunov learning and the set $S \subseteq \mathcal{R}(\gamma)$ is a finite subset of points in the valid region. We assume that the loss function is differentiable, and consequently training follows a sequence of gradient update steps $(\theta', \beta') = (\theta, \beta) - \mu \sum_{i \in S} \nabla_{\theta, \beta} \mathcal{L}(x_i; V_\theta, \pi_\beta)$.

Clearly, the choice of the set $S$ is pivotal to successfully learning $V_\theta$ and $\pi_\beta$ with provable stability properties. Prior approaches for learning Lyapunov functions represented by neural networks use one of two ways to generate $S$. The first is to generate a fixed set $S$ comprised of uniformly random samples from the domain [Chang and Gao, 2021, Richards et al., 2018]. However, this fails to learn verifiable Lyapunov functions. In the second, $S$ is not fixed, but changes in each learning iteration, starting with randomly generated samples in the domain, and then using the verification tool, such as dReal, in each step of the learning process to update $S$ [Chang et al., 2019, Zhou et al., 2022]. However, verification is often slow, and becomes a significant bottleneck in the learning process.

Our key idea is to combine the benefits of these two ideas with a fast gradient-based heuristic approach for generating counterexamples. In particular, our proposed approach for training Lyapunov control functions and associated control policies (Algorithm 1; DITL) involves five parts:

1. heuristic counterexample generation (lines 10-12),
2. choosing a collection $S$ of inputs to update $\theta$ and $\beta$ in each training (gradient update) iteration (lines 9-13),
3. the design of the loss function $\mathcal{L}$,
4. initialization of policy $\pi_\beta$ (line 3), and
5. warm starting the training (line 4).

We describe each of these next.

---

**Algorithm 1** DITL Lyapunov learning algorithm.

---

 1: **Input:** Dynamical system model $f(x, u)$ and target valid region $\mathcal{R}(\gamma)$
 2: **Output:** Lyapunov function $V_\theta$ and control policy $\pi_\beta$
 3: $\pi_\beta =$ Initialize()
 4: $V_\theta =$ PreTrainLyapunov($N_0, \pi_\beta$)
 5: $B =$ InitializeBuffer()
 6: $W \leftarrow \emptyset$
 7: **while** True **do**
 8:   **for** $N$ iterations **do**
 9:     $\hat{S} =$ Sample($r, B$) //sample $r$ points from $B$
10:     $T =$ Sample($q, \mathcal{R}(\gamma)$) //sample $q$ points from $\mathcal{R}(\gamma)$
11:     $T' =$ PGD($T$)
12:     $T'' =$ FilterCounterExamples($T'$)
13:     $S = \hat{S} \cup T'' \cup W$
14:     $B = B \cup T''$
15:     $(\theta, \beta) \leftarrow (\theta, \beta) - \mu \sum_{i \in S} \nabla_{\theta, \beta} \mathcal{L}(x_i; V_\theta, \pi_\beta)$
16:   **end for**
17:   (success, $\hat{W}$) = Verify($V_\theta, \pi_\beta$)
18:   **if** success **then**
19:     **return** $V_\theta, \pi_\beta$
20:   **else**
21:     $W \leftarrow W \cup \hat{W}$
22:   **end if**
23: **end while**
24: **return** FAILED // Timeout

---

**Heuristic Counterexample Generation** As discussed in Section 3, verification in general involves two optimization problems: $\min_{x \in \mathcal{R}(\gamma)} V_\theta(x) - V_\theta(f(x, \pi_\beta(x)))$ and $\min_{x \in \mathcal{R}(\gamma)} V_\theta(x)$. We propose a *projected gradient descent* (PGD) approach for approximating solutions to either of these problems. PGD proceeds as follows, using the second minimization problem as an illustration; the process is identical in the first case. Beginning with a starting point $x_0 \in \mathcal{R}(\gamma)$, we iteratively update $x_{k+1}$:

$$x_{k+1} = \Pi\{x_k + \alpha_k \text{sgn}(\nabla V_\theta(x_k))\},$$

where $\Pi\{\}$ projects the gradient update step onto the domain $\mathcal{R}(\gamma)$ by clipping each dimension of $x_k$, and $\alpha_k$ is the PGD learning rate. Note that as $f$ is typically differentiable, we can take gradients directly through it when we apply PGD for the first verification problem above.

We run this procedure for $K$ iterations, and return the result (which may or may not be an actual counterexample). Let $T' =$ PGD($T$) denote running PGD for a set of $T$ starting points for both of the optimization problems above, resulting in a corresponding set $T'$ of counterexamples.

**Selecting Inputs for Gradient Updates** A natural idea for selecting samples $S$ is to simply add counterexamples identified in each iteration of the Lyapunov learning process. However, as $S$ then grows without bound, this progressively slows down the learning process. In addition, it is important for $S$ to contain a relatively broad sample of the valid region to ensure that counterexamples we generate along the way do not cause undesired distortion of $V_\theta$ and $\pi_\beta$ in subregions not well represented in $S$. Finally, we need to retain the memory of previously generated counterexamples, as

these are often particularly challenging parts of the input domain for learning. We therefore construct $S$ as follows.

We create an input buffer $B$, and initialize it with a random sample of inputs from $\mathcal{R}(\gamma)$. Let $W$ denote a set of counterexamples that we wish to remember through the entire training process, initialized to be empty. In our implementation, $W$ consists of all counterexamples generated by the sound verification tools described in Section 3.

At the beginning of each training update iteration, we randomly sample a fixed-size subset $\hat{S}$ of the buffer $B$. Next, we take a collection $T$ of random samples from $\mathcal{R}$ to initialize PGD, and generate $T' = \text{PGD}(T)$. We then filter out all the counterexamples from $T'$, retaining only those with either $V_\theta(x) \le 0$ or $V_\theta(x) - V_\theta(f(x, \pi_\beta(x))) \le \epsilon$, where $\epsilon$ is a non-negative hyperparameter; this yields a set $T''$. We then use $S = \hat{S} \cup T'' \cup W$ for the gradient update in the current iteration. Finally, we update the buffer $B$ by adding to it $T''$. This process is repeated for $N$ iterations.

After $N$ iterations, we check to see if we have successfully learned a stable policy and a Lyapunov control function by using the tools from Section 3 to verify $V_\theta$ and $\pi_\beta$. If verification succeeds, training is complete. Otherwise, we use the counterexamples $\hat{W}$ generated by the MILP solver to update $W = W \cup \hat{W}$, and repeat the learning process above.

**Loss Function Design**  Next, we design of the loss function $\mathcal{L}(x; V_\theta, \pi_\beta)$ for a given input $x$. A key ingredient in this loss function is a term that incentivizes the learning process to satisfy condition (b) of $\epsilon$-stability: $\mathcal{L}_1(x; V_\theta, \pi_\beta) = \text{ReLU}(V_\theta(f(x, \pi_\beta(x))) - V_\theta(x) + \eta)$, where the parameter $\eta \ge 0$ determines how aggressively we try to satisfy this condition during learning.

There are several options for learning $V_\theta$ satisfying conditions (a) and (c) of $\epsilon$-stability. The simplest is to set all bias terms in the neural network to zero, which immediately satisfies $V_\theta(0) = 0$. An effective way to deal with condition (c) is to maximize the lower bound on $V_\theta(x)$. To this end, we propose to make use of the following loss term: $\mathcal{L}_2(x; V_\theta, \pi_\beta) = \text{ReLU}(-V_\theta^{LB})$, where $V_\theta^{LB}$ is the lower bound on the Lyapunov function over the target domain $\mathcal{R}(\gamma)$. We use use $\alpha, \beta$-CROWN [Xu et al., 2020] to obtain this lower bound.

The downside of setting bias terms to zero is that we lose many learnable parameters, reducing flexibility of the neural network. If we consider a general neural network, on the other hand, it is no longer the case that $V_\theta(x) = 0$ by construction. However, it is straigthforward to ensure this by defining the final Lyapunov function as $\tilde{V}_\theta(x) = V_\theta(x) - V_\theta(0)$. Now, satisfying condition (c) amounts to satisfying the condition that $V_\theta(x) \ge V_\theta(0)$, which we accomplish via the following pair of loss terms: $\mathcal{L}_3(x; V_\theta, \pi_\beta) = \text{ReLU}(V_\theta(0) - V_\theta(x) + \mu \min(\|x\|_2, \nu))$, where $\mu$ and $\nu$ are hyperparameters of the term $\mu \min(\|x\|_2, \nu)$ which effectively penalizes $V_\theta(x)$ for being too large, and $\mathcal{L}_4(x; V_\theta, \pi_\beta) = \|V_\theta(0)\|_2^2$.

The general loss function is then a weighted sum of these loss terms,
$$\mathcal{L}(x; V_\theta, \pi_\beta) = \mathcal{L}_1(x; V_\theta, \pi_\beta) + c_2 \mathcal{L}_2(x; V_\theta, \pi_\beta) + c_3 \mathcal{L}_3(x; V_\theta, \pi_\beta) + c_4 \mathcal{L}_4(x; V_\theta, \pi_\beta).$$
When we set bias terms to zero, we would set $c_3 = c_4 = 0$; otherwise, we set $c_2 = 0$.

**Initialization**  We consider two approaches for initializing the policy $\pi_\beta$. The first is to linearize the dynamics $f$ around the origin, and use a policy computing based on a linear quadratic regulator (LQR) to obtain a simple linear policy $\pi_\beta$. The second approach is to use deep reinforcement, such as PPO [Liu et al., 2019], where rewards correspond to stability (e.g., reward is the negative value of the $l_2$ distance of state to origin). To initialize $V_\theta$, we fix $\pi_\beta$ after its initialization, and follow our learning procedure above using solely heuristic counterexample generation to pre-train $V_\theta$.

The next result now follows by construction.

**Theorem 4.1.** *If Algorithm 1 returns $V_\theta, \pi_\beta$, they are guaranteed to satisfy $\epsilon$-stability conditions.*

## 5  Experiments

### 5.1  Experiment Setup

**Benchmark Domains**  Our evaluation of the proposed DITL approach uses four benchmark control domains: *inverted pendulum*, *path tracking*, *cartpole*, and *drone planar vertical takeoff and landing (PVTOL)*. Details about these domains are provided in the Supplement.

**Baselines** We compare the proposed approach to four baselines: *linear quadratic regulator (LQR)*, *sum-of-squares (SOS)*, *neural Lyapunov control (NLC)* [Chang et al., 2019], and *neural Lyapunov control for unknown systems (UNL)* [Zhou et al., 2022]. The first two, LQR and SOS, are the more traditional approaches for computing Lyapunov functions when the dynamics are either linear (LQR) or polynomial (SOS) [Tedrake, 2009]. LQR solutions (when the system can be stabilized) are obtained through matrix multiplication, while SOS entails solving a semidefinite program (we solve it using YALMIP with MOSEK solver in MATLAB 2022b). The next two baselines, NLC and UNL, are recent approaches for learning Lyapunov functions in continuous-time systems (no approach for learning provably stable control using neural network representations exists for discrete time systems). Both NLC and UNL yield provably stable control for general non-linear dynamical systems, and are thus the most competitive baselines to date. In addition to these baselines, we consider an ablation in which PGD-based counterexample generation for our approach is entirely replaced by the sound MILP-based method during learning (we refer to it as DITL-MILP).

**Efficacy Metrics** We compare approaches in terms of three efficacy metrics. The first is (serial) runtime (that is, if no parallelism is used), which we measure as wall clock time when only a single task is running on a machine. For inverted pendulum and path tracking, all comparisons were performed on a machine with AMD Ryzen 9 5900X 12-Core Processor and Linux Ubuntu 20.04.5 LTS OS. All cartpole and PVTOL experiments were run on a machine with a Xeon Gold 6150 CPU (64-bit 18-core x86), Rocky Linux 8.6. UNL and RL training for Path Tracking are the only two cases that make use of GPUs, and was run on NVIDIA GeForce RTX 3090. The second metric was the size of the valid region, measured using $\ell_2$ norm for NLC and UNL, and $\ell_\infty$ norm (which dominates $\ell_2$) for LQR, SOS, and our approach. The third metric is the *region of attraction (ROA)*. Whenever verification fails, we set ROA to 0. Finally, we compare all methods whose results are stochastic (NLC, UNL, and ours) in terms of *success rate*.

**Verification Details** For LQR, SOS, NLC, and UNL, we used dReal as the verification tool, as done by Chang et al. [2019] and Zhou et al. [2022]. For DITL verification we used CPLEX version 22.1.0.

## 5.2  Results

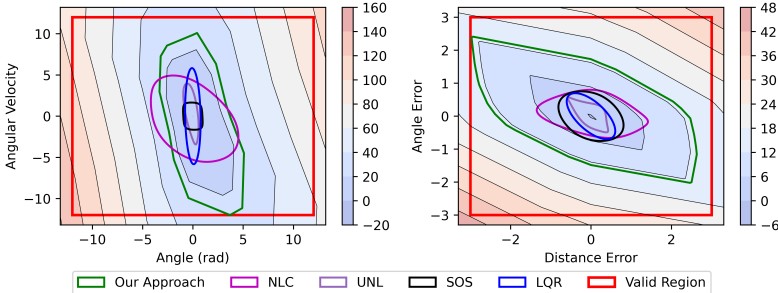

Figure 1: ROA plot of inverted pendulum (left) and path tracking (right). We select the best result for each method.

**Inverted Pendulum** For the inverted pendulum domain, we initialize the control policy using the LQR solution (see the Supplement for details). We train $V_\theta$ with non-zero bias terms. We set $\epsilon = 0.1$ ($< 0.007\%$ of the valid region), and approximate ROA using a grid of 2000 cells along each coordinate using the level set certified with MILP (4). All runtime is capped at 600 seconds, except the UNL baseline, which we cap at 16 minutes, as it tends to be considerably slower than other methods. Our results are presented in Table 1. While LQR and SOS are fastest, our approach (DITL) is the next fastest, taking on average $\sim 8$ seconds, with NLC and UNL considerably slower. However, DITL yields an ROA *a factor >4 larger* than the nearest baseline (LQR), and 100% success rate. Finally, the DITL-MILP ablation is two orders of magnitude slower (runtime $> 300$ seconds) *and* less effective (average ROA=42, 80% success rate) than DITL. We visualize maximum ROA produced by all methods in Figure 1 (left).

**Path Tracking** In path tracking, we initialize our approach using both the RL and LQR solutions, drawing a direct comparison between the two (see the Supplement for details). We set $\epsilon = 0.1$. The running time of RL is $\sim 155$ seconds. The results are provided in Table 2. We can observe that both RL- and LQR-initialized variants of our approach outperform all prior art, with RL exhibiting *a*

Table 1: Inverted Pendulum

| | Valid Region | Runtime (s) | ROA | Max ROA | Success Rate |
|---|---|---|---|---|---|
| NLC (free) | $\|x\|_2 \leq 6.0$ | $28 \pm 29$ | $11 \pm 4.6$ | 22 | 100% |
| NLC (max torque 6.0) | $\|x\|_2 \leq 6.0$ | $519 \pm 184$ | $13 \pm 27$ | 66 | 20% |
| UNL (max torque 6.0) | $\|x\|_2 \leq 4.0$ | $821 \pm 227$ | $1 \pm 2$ | 7 | 30% |
| LQR | $\|x\|_\infty \leq 5.8$ | $< 1$ | 14 | 14 | success |
| SOS | $\|x\|_\infty \leq 1.7$ | $< 1$ | 6 | 6 | success |
| DITL | $\|x\|_\infty \leq 12$ | $8.1 \pm 4.7$ | $\mathbf{61 \pm 31}$ | **123** | 100% |

Table 2: Path Tracking

| | Valid Region | Runtime (s) | ROA | Max ROA | Success Rate |
|---|---|---|---|---|---|
| NLC | $\|x\|_2 \leq 1.0$ | $109 \pm 81$ | $0.5 \pm 0.2$ | 0.76 | 100% |
| NLC | $\|x\|_2 \leq 1.5$ | $151 \pm 238$ | $1.4 \pm 0.9$ | 2.8 | 80% |
| UNL | $\|x\|_2 \leq 0.8$ | $925 \pm 110$ | $0.1 \pm 0.2$ | 0.56 | 10% |
| LQR | $\|x\|_\infty \leq 0.7$ | $< 1$ | 1.02 | 1.02 | success |
| SOS | $\|x\|_\infty \leq 0.8$ | $< 1$ | 1.8 | 1.8 | success |
| DITL (LQR) | $\|x\|_\infty \leq 3.0$ | $9.8 \pm 4$ | $8 \pm 3$ | 12.5 | 100% |
| DITL (RL) | $\|x\|_\infty \leq 3.0$ | $14 \pm 11$ | $\mathbf{9 \pm 3.5}$ | **16** | 100% |

*factor of 5 advantage* over the next best (SOS, in this case) in terms of ROA, and nearly a factor of 6 advantage in terms of maximum achieved ROA (NLC is the next best in this case). Moreover, our approach again has a 100% success rate. Our runtime is an order of magnitude lower than NLC or UNL. Overall, the RL-initialized variant slightly outperforms LQR initialization. The DITL-MILP ablation again performs far worse than DITL: running time is several orders of magnitude slower (at $> 550$ seconds), with low efficacy (ROA is 1.1, success rate 10%). We visualize comparison of maximum ROA produced by all methods in Figure 1 (right).

**Cartpole** For cartpole, we used LQR for initialization, and set bias terms of $V_\theta$ to zero. We set $\epsilon = 0.1$ (0.01% of the valid region area) and the running time limit to 2 hours for all approaches. None of the baselines successfully attained a provably stable control policy and associated Lyapunov function for this problem. Failure was either because we could find counterexamples within the target valid region, or because verification exceeded the time limit. DITL found a valid region of $\|x\|_\infty \leq 1.0$, in $\leq 1.6$ hours with a 100% success rate, and average ROA of $0.021 \pm 0.012$.

**PVTOL** The PVTOL setup was similar to cartpole. We set $\epsilon = 0.1$ (0.0001% of the valid region area), and maximum running time to 24 hours. None of the baselines successfully identified a provably stable control policy. In contrast, DITL found one within $13 \pm 6$ hours on average, yielding a 100% success rate. We identified a valid region of $\|x\|_\infty \leq 1.0$, and ROA of $0.011 \pm 0.008$.

# 6 Conclusion

We presented a novel algorithmic framework for learning Lyapunov control functions and policies for discrete-time nonlinear dynamical systems. Our approach combines mixed-integer linear programming verification tool with a training procedure that leverages gradient-based approximate verification. This combination enables a significant improvement in effectiveness compared to prior art: our experiments demonstrate that our approach yields several factors larger regions of attraction in inverted pendulum and path tracking, and ours is the only approach that successfully finds stable policies in cartpole and PVTOL.

## Acknowledgments

This research was partially supported by the NSF (grants CNS-1941670, ECCS-2020289, IIS-1905558, IIS-2214141, and CNS-2231257), AFOSR (grant FA9550-22-1-0054), ARO (grant W911NF-19-1-0241), and NVIDIA.

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

# A Further Details about Experiments

## A.1 Dynamics

Since these are described as continuous-time domains, we set the time between state updates to be $0.05s$ in all cases for our discretized versions of these systems.

**Inverted Pendulum** We use the following standard dynamics model for inverted pendulum [Chang et al., 2019, Richards et al., 2018, Zhou et al., 2022]:

$$\ddot{\theta} = \frac{mg\ell\sin(\theta) + u - b\dot{\theta}}{m\ell^2},$$

with gravity constant $g = 9.81$, ball weight $m = 0.15$, friction $b = 0.1$, and pendulum length $l = 0.5$. We set the maximum torque allowed to be $6Nm$, meaning $u \in [-6.0, 6.0]$, and it is applied to all experiments except for the NLC (free) case.

**Path Tracking** Our *path tracking* dynamics follows Snider [2009] and Chang et al. [2019]:

$$\dot{s} = \frac{v\cos(\theta_e)}{1 - \dot{e}_{ra}\kappa(s)}, \quad \dot{e}_{ra} = v\sin(\theta_e), \dot{\theta}_e = \frac{v\tan(\delta)}{L} - \frac{v\kappa(s)\cos(\theta_e)}{1 - \dot{e}_{ra}\kappa(s)},$$

where the control policy determines the steering angle $\delta$. $e_{ra}$ is the distance error and $\theta_e$ is the angle error. We can simplify the dynamics by defining $u = \tan(\delta)$, so that non-linearity of dynamics is only in terms of state variables. Since in practice $|\delta| \leq 40°$, we set $u \in [\tan(-40°), \tan(40°)]$; this is applied to all experiments. We set speed $v = 2.0$, while the track is a circle with radius 10.0 (and, thus, $\kappa = 0.1$) and $L = 1.0$.

For RL in path tracking, the reward function is $-\frac{1}{10}(e_{ra}^2 + \theta_e^2)^{1/2}$, and we use PPO [Schulman et al., 2017], training over 50K steps.

**Cartpole** For *cartpole* we use the dynamics from Tedrake [2009]. The original dynamics is

$$\ddot{x} = \frac{1}{m_c + m_p\sin^2\theta}\left[f_x + m_p\sin\theta(l\dot{\theta}^2 + g\cos\theta)\right] \tag{6}$$

$$\ddot{\theta} = \frac{1}{l(m_c + m_p\sin^2\theta)}\left[-f_x\cos\theta - m_pl\dot{\theta}^2\cos\theta\sin\theta - (m_c + m_p)g\sin\theta\right]. \tag{7}$$

We change the variable and rename $\pi - \theta$ as $\theta$, so that in our setting, $\theta$ represents the angle between the pole and the upward horizontal direction, the pole angle is positive if it is to the right. The purpose of this transformation is that the equilibrium point is the origin.

Using the second order chain rule

$$\frac{d^2y}{dz^2} = \frac{d^2y}{du^2}\left(\frac{du}{dz}\right)^2 + \frac{dy}{du}\frac{d^2u}{dz^2}$$

let $y = \pi - \theta, z = t, u = \theta$, we have the new dynamics as

$$\ddot{x} = \frac{1}{m_c + m_p\sin^2\theta}\left[f_x + m_p\sin\theta(l\dot{\theta}^2 - g\cos\theta)\right] \tag{8}$$

$$\ddot{\theta} = \frac{1}{l(m_c + m_p\sin^2\theta)}\left[-f_x\cos\theta - m_pl\dot{\theta}^2\cos\theta\sin\theta + (m_c + m_p)g\sin\theta\right]. \tag{9}$$

Here $m_c = 1.0$ is the weight of the cart, $m_p = 0.1$ is the weight of the pole, $x$ is the horizontal position of the cart, $l = 1.0$ is the length of the pole, $g = 9.81$ is the gravity constant and $f_x$ is the controller (force applied to the cart). Furthermore, the maximum force allowed to the cart is set to be $30N$, meaning $f_x \in [-30, 30]$; this is applied to all experiments.

**PVTOL**    For *PVTOL*, our dynamics follows Singh et al. [2021]. Specifically, the system has the following state representation: $x = \left(p_x, p_z, v_x, v_z, \phi, \dot{\phi}\right)$. $(p_x, p_z)$ and $(v_x, v_z)$ are the $2D$ position and velocity, respectively, and $(\phi, \dot{\phi})$ are the roll and angular rate. Control $u \in \mathbb{R}^2_{>0}$ corresponds to the controlled motor thrusts. The dynamics $f(x, u)$ are described by

$$
\dot{x}(t) =
\begin{bmatrix}
v_x \cos\phi - v_z \sin\phi \\
v_x \sin\phi + v_z \cos\phi \\
\dot{\phi} \\
v_z \dot{\phi} - g \sin\phi \\
-v_x \dot{\phi} - g \cos\phi \\
0
\end{bmatrix}
+
\begin{bmatrix}
0 & 0 \\
0 & 0 \\
0 & 0 \\
0 & 0 \\
(1/m) & (1/m) \\
l/J & (-l/J)
\end{bmatrix}
u
$$

where $g = 9.8$ is the acceleration due to gravity, $m = 4.0$ is the mass, $l = 0.25$ is the moment-arm of the thrusters, and $J = 0.0475$ is the moment of inertia about the roll axis. The maximum force allowed to both $u_1$ and $u_2$ is set to be $39.2N$, which is $mg$, meaning $u_1, u_2 \in [0, 39.2]$; this is applied to all experiments.

## A.2    Computation of the LQR Solution

Given the system $dx/dt = Ax + Bu$, the objective function of the LQR is to compute the optimal controller $u = -Kx$ that minimizes the quadratic cost $\int_0^\infty (x'Qx + u'Ru)dt$, where $R$ and $Q$ are identity matrices. The details of the system linearization as well as the LQR solution are described below.

**Inverted Pendulum**    We linearize the system as

$$
A = \begin{bmatrix} 0 & 1 \\ \frac{g}{l} & -\frac{b}{ml^2} \end{bmatrix}, B = \begin{bmatrix} 0 \\ \frac{1}{ml^2} \end{bmatrix}
$$

The final LQR solution is $u = -1.977252234\theta - 0.976240064\dot{\theta}$.

**Path Tracking**    We linearize the system as

$$
A = \begin{bmatrix} 0 & 2 \\ 0 & -0.04 \end{bmatrix}, B = \begin{bmatrix} 0 \\ 1 \end{bmatrix}.
$$

The final LQR solution is $u' = 2u + 0.1 = -e_{ra} - 2.19642572\theta_e$.

**Cartpole**    We linearize the system as

$$
A = \begin{bmatrix} 0 & 1 & 0 & 0 \\ 0 & 0 & -0.98 & 0 \\ 0 & 0 & 0 & 1 \\ 0 & 0 & 10.78 & 0 \end{bmatrix}, B = \begin{bmatrix} 0 \\ 1 \\ 0 \\ -1 \end{bmatrix}.
$$

The final LQR solution is $u = x + 2.4109461\dot{x} + 34.36203947\theta + 10.70094483\dot{\theta}$.

**PVTOL**    We linearize the system as

$$
A = \begin{bmatrix}
0 & 0 & 0 & 1 & 0 & 0 \\
0 & 0 & 0 & 0 & 1 & 0 \\
0 & 0 & 0 & 0 & 0 & 1 \\
0 & 0 & -g & 0 & 0 & 0 \\
0 & 0 & 0 & 0 & 0 & 0 \\
0 & 0 & 0 & 0 & 0 & 0
\end{bmatrix}, B = \begin{bmatrix}
0 & 0 \\
0 & 0 \\
0 & 0 \\
0 & 0 \\
\frac{1}{m} & \frac{1}{m} \\
\frac{r}{J} & -\frac{r}{J}
\end{bmatrix}
$$

The final LQR solution is $u_1 = 0.70710678x - 0.70710678y - 5.039548710 + 1.10781077\dot{x} - 1.82439774\dot{y} - 1.20727555\dot{\theta}$, and $u_2 = -0.70710678x - 0.70710678y + 5.039548710 - 1.10781077\dot{x} - 1.82439774\dot{y} + 1.20727555\dot{\theta}$.

### A.3 PPO Training for Path Tracking

We use the implementation in Stable-Baselines [Raffin, 2020] for PPO training. The hyperparameters of are fine-tuned using their auto-tuning script with a budget of 1000 trials with a maximum of 50000 steps. The final hyperparameters are in Table 3.

Table 3: Hyperparameter for PPO (Path Tracking)

| Parameter | Value |
|---|---|
| policy | MlpPolicy |
| n_timesteps | !!float 100000 |
| batch_size | 32 |
| n_steps | 64 |
| gamma | 0.95 |
| learning_rate | 0.000208815 |
| ent_coef | 1.90E-06 |
| clip_range | 0.1 |
| n_epochs | 10 |
| gae_lambda | 0.99 |
| max_grad_norm | 0.8 |
| vf_coef | 0.550970466 |
| activation_fn | nn.ReLU |
| log_std_init | -0.338380542 |
| ortho_init | False |
| net_arch | [8, 8] |
| vf | [8, 8] |
| sde_sample_freq | 128 |

### A.4 Benchmark Models

When relevant, benchmark models (as well as our approach) are run for 10 seeds (seed 0 to 9). Mean $\pm$ standard deviation are reported in Table 1 and Table 2. For SOS benchmark, we use polynomials of degree $\leq 6$ for all environments.

**NLC and UNL** For Cartpole and PVTOL, we train against diameter 1.0 under the $l_2$ norm for NLC and UNL. The main reason for the failure of training is that the dReal verifier becomes incredibly show after some iterations. For example, for seed 0, NLC only finished 167 certifications within 2 hours limit for Cartpole, and 394 certifications within 24 hours limit for PVTOL.

**LQR and SOS** For Cartpole environment, we first certify against target region $0.1 \leq ||x||_\infty \leq 0.16$ for LQR benchmark where dReal returns the counterexample; we later attempt to certify against $0.1 \leq ||x||_\infty \leq 0.15$, dReal did not return any result within the 2-hour limit. For SOS benchmark, dReal verifier did not return any result within the 2 hours limit against target region $0.1 \leq ||x||_\infty \leq 0.5$. For PVTOL environment, LQR benchmark returns the counterexample against target region $0.1 \leq ||x||_\infty \leq 0.5$ after 10 hours, we then attempt to certify against $0.1 \leq ||x||_\infty \leq 0.4$ and fail to return any result within the remaining 14 hours time limit. For SOS benchmark, dReal verifier did not return any result within the 24 hours limit against target region $0.1 \leq ||x||_\infty \leq 0.5$.

### A.5 Bounding the Functions

There are three types of bounding in our paper: 1) bound the dynamics $f(x, u)$ with linear upper/lower bound functions for verification; 2) bound the dynamics with constants for the calculation of $M$ for ReLU; 3) bound the dynamics with constants for the calculation of ROA. Note that the bounding for 2) and 3) are similar, except that for 2) it is calculated for each subgrid, and for 3) it is calculated for the entire valid region.

**Inverted Pendulum**   Split the region: we split the region over the domain of $\theta$, please refer to our code for the details of the splitting.

(1) Bound for verification: since the only non-linear part of the function is $\sin(\theta)$, we use the upper/lower bounds in $\alpha, \beta$-CROWN. (2) Bound for $M$: $|\ddot{\theta}| \leq \frac{g}{l} + \frac{|u|_{\max}}{ml^2} + \frac{b|\dot{\theta}|_{\max}}{ml^2}$, where $|u|_{\max}$ is calculated using IBP bound. (3) Bound for ROA: $||f(x,u)||_\infty = \gamma + \max(\gamma, \frac{mgl+|u|_{\max}+b\gamma}{ml^2})dt$, where $|u|_{\max}$ is the maximum force allowed.

**Path Tracking**   Split the region: we split the region over the domain of $\theta_e$, please refer to our code for the details of the splitting.

(1) Bound for verification: for $\dot{e}_{ra}$ since the only nonlinear function is $\sin(\theta)$, we use the bound in $\alpha, \beta$-CROWN; for $\dot{\theta}_e$ we bound the term $\frac{v\kappa(s)\cos(\theta_e)}{1-\dot{e}_{ra}\kappa(s)}$, which is concave in interval $[-\pi/2, \pi/2]$, and convex otherwise. (2) Bound for $M$: $|\dot{e}_{ra}| \leq v$, $|\dot{\theta}_e| \leq \frac{v|u|_{\max}}{L} + \frac{v}{R-v}$, where $|u|_{\max}$ is calculated using IBP bound. (3) Bound for ROA: $||f(x,u)||_\infty = \gamma + \max(v, \frac{v|u|_{\max}}{L} + \frac{v}{R-v})dt$, where $|u|_{\max}$ is the maximum force allowed.

**Cartpole**   Split the region: we split the region over the domain of $\theta$ and $\dot{\theta}$ with interval 0.1 when either $\theta$ or $\dot{\theta}$ is greater than 0.1. In case when both $\theta$ and $\dot{\theta}$ are smaller than 0.1, we split $\theta, \dot{\theta}$ into intervals of 0.05. Please refer to our code for the details of the splitting.

(1) Bound for verification: we split the dynamics into six functions and bound them separately for both one variable and multiple variables: $f_1 = \frac{1}{m_c+m_p \sin^2 \theta}$, $f_2 = \frac{-\cos\theta}{l(m_c+m_p \sin^2 \theta)}$, $f_3 = \frac{-gm_p \sin\theta \cos\theta}{m_c+m_p \sin^2 \theta}$, $f_4 = \frac{(m_c+m_p)g\sin\theta}{l(m_c+m_p \sin^2 \theta)}$, $f_5 = \frac{m_p l \sin\theta \dot{\theta}^2}{m_c+m_p \sin^2 \theta}$, $f_6 = \frac{-m_p l \dot{\theta}^2 \cos\theta \sin\theta}{l(m_c+m_p \sin^2 \theta)}$. (2) Bound for $M$: $|\ddot{x}| \leq |u|_{\max} + m_p l|\dot{\theta}|^2_{\max} + \frac{m_p g}{2}$, $|\ddot{\theta}| \leq \frac{1}{l}(|u|_{\max} + \frac{m_p l|\dot{\theta}|^2_{\max}}{2} + (m_c+m_p)g)$ where $|u|_{\max}$ is calculated using IBP bound. (3) Bound for ROA: $||f(x,u)||_\infty = \gamma + \max(\gamma, |u|_{\max} + m_p l\gamma^2 + \frac{m_p g}{2}, \frac{1}{l}(|u|_{\max} + \frac{m_p l\gamma^2}{2} + (m_c+m_p)g))dt$, where $|u|_{\max}$ is the maximum force allowed.

**PVTOL**   Split the region: we split the region over the domain of $\theta$ with interval 0.25 and split the region over the domain of $\dot{x}, \dot{y}, \dot{\theta}$ with interval 0.5. Furthermore, in the MILP we add a constraint to ensure the $l_\infty$ norm of state value is no less than 0.1. Please refer to our code for the details of the splitting.

(1) Bound for verification: we split the dynamics into three functions and bound them separately for multi-variables: $f_1 = x\cos y$, $f_2 = x\sin y$, $f_3 = xy$. (2) Bound for $M$: we bound each term of the dynamics with the min/max values using the monotonic property. (3) Bound for ROA: $||f(x,u)||_\infty = \gamma + \max(\gamma, \gamma^2 + g + \frac{2|u|_{\max}}{m}, 2\gamma, \frac{2l|u|_{\max}}{J})dt$, where $|u|_{\max}$ is the maximum force allowed.

## A.6   Speeding up Verification

When there is a limit on parallel resources, so that verification must be done serially for the subgrids $k$, we can reduce the number of grids to verify in each step as follows. The first time a verifier is called, we perform it over all subgrids; generally, only a subset will return a counterexample. Subsequently, we only verify these subgrids until none return a counterexample, and only then attempt full verification.

## A.7   ROA Calculation

For Inverted Pendulum and Path Tracking, we approximate ROA using a grid of 2000 cells along each coordinate. For Cartpole, we use a grid of 150 cells along each coordinate, and for PVTOL, we use a grid of 50 cells along each coordinate.

For our baselines where the systems are continuous, the set levels are $\{x \in \mathcal{R} \mid V(x) < \rho\}$ where $\rho = \min_{x \in \partial\mathcal{R}} V(x)$. Note that for discrete-time systems, having $\rho = \min_{x \in \partial\mathcal{R}} V(x)$ is not sufficient. For example, assume $\mathcal{R}$ is the valid region, any point outside of $\mathcal{R}$ satisfy $V(f(x,u)) > V(x)$, and $\forall x \in \partial\mathcal{R}, V(x) = \rho$. Assume we have $x \in \mathcal{R} \setminus \partial\mathcal{R}$ and $f(x,u) = y, y \notin \mathcal{R}$ (this is possible since

the system is discrete). Since $x$ is in the valid region, we have $V(y) < V(x) < \rho$. However, since $y$ is no longer within the valid region, assume $V(f(y, u)) > \rho > V(y)$, the system will not converge to equilibrium point starting from point $y$. This means $\{x \in \mathcal{R} \mid V(x) < \rho\}$ is not a subset of ROA.

## B    Computing Linear Bounds on Lipschitz-Continuous Dynamics

**Theorem B.1.** *Given function $g(x) : \mathbb{R}^n \mapsto \mathbb{R}$ where the Lipschitz constant is $\lambda$, we have $g(x) \leq g(x_{1,l}, x_{2,l}, \ldots, x_{n,l}) + \lambda \cdot \sum_i (x_i - x_{i,l})$ and $g(x_{1,l}, x_{2,l}, \ldots, x_{n,l}) - \lambda \cdot \sum_i (x_i - x_{i,l}) \leq g(x)$, where $x_i \in [x_{i,l}, x_{i,u}], \forall i \in [n]$.*

*Proof.* For any $i \in [n]$, given $x_i \in [x_{i,l}, x_{i,u}]$, for arbitrary fixed values along dimension $[n] \setminus \{j\}$ (we denote them as $x_{-i}$), we have $|g(x_i, x_{-i}) - g(x_{i,l}, x_{-i})| \leq \lambda |x_i - x_{i,l}|$.

- we show that $g(x_i, x_{-i}) \leq g(x_{i,l}, x_{-i}) + \lambda(x_i - x_{i,l})$
    - if $g(x_i, x_{-i}) \geq g(x_{i,l}, x_{-i})$, then $g(x_i, x_{-i}) - g(x_{i,l}, x_{-i}) \leq \lambda(x_i - x_{i,l})$, meaning $g(x_i, x_{-i}) \leq g(x_{i,l}, x_{-i}) + \lambda(x_i - x_{i,l})$.
    - if $g(x_i, x_{-i}) < g(x_{i,l}, x_{-i})$, since $\lambda(x_i - x_{i,l}) \geq 0$, we have $g(x_i, x_{-i}) \leq g(x_{i,l}, x_{-i}) + \lambda(x_i - x_{i,l})$.

- we show that $g(x_{i,l}, x_{-i}) - \lambda(x_i - x_{i,l}) \leq g(x_i, x_{-i})$
    - if $g(x_i, x_{-i}) \geq g(x_{i,l}, x_{-i})$, then since $-\lambda(x_i - x_{i,l}) \leq 0$, we have $g(x_{i,l}, x_{-i}) - \lambda(x_i - x_{i,l}) \leq g(x_i, x_{-i})$.
    - if $g(x_i, x_{-i}) < g(x_{i,l}, x_{-i})$, then $g(x_{i,l}, x_{-i}) - g(x_i, x_{-i}) \leq \lambda(x_i - x_{i,l})$, meaning $g(x_{i,l}, x_{-i}) - \lambda(x_i - x_{i,l}) \leq g(x_i, x_{-i})$.

By applying the above result along all $n$ dimensions recursively, we have the upper bound as $g(x) \leq g(x_{1,l}, x_{2,l}, \ldots, x_{n,l}) + \sum_i \lambda(x_i - x_{i,l})$, and the lower bound as $g(x_{1,l}, x_{2,l}, \ldots, x_{n,l}) - \sum_i \lambda(x_i - x_{i,l}) \leq g(x)$. □

## C    Ablation Study

In this section, we conduct an ablation study on the MILP solver. Table 4 are the results for DITL-dReal, in which we use our framework, but replace all the MILP components with dReal. As we can see, using MILP is an essential ingredient in the success of the proposed approach:

Table 4: Ablation study on MILP solver.

| Environment | Runtime | ROA | Max ROA | Success Rate |
|---|---|---|---|---|
| Inverted Pendulum (DITL-dReal) | **6.0 ± 1.7 (s)** | $57 \pm 24$ | 75 | **100%** |
| **Inverted Pendulum (DITL, main paper)** | 8.1 ± 4.7(s) | **61 ± 31** | **123** | **100%** |
| Path Tracking (RL, DITL-dReal) | $600 \pm 0$ (s) | $0 \pm 0$ | 0 | 0% |
| **Path Tracking (RL, DITL, main paper)** | **14 ± 11 (s)** | **9 ± 3.5** | **16** | **100%** |
| Path Tracking (LQR, DITL-dReal) | $420.9 \pm 182$ (s) | $4 \pm 4$ | 11 | 60% |
| **Path Tracking (LQR, DITL, main paper)** | **9.8 ± 4 (s)** | **8 ± 3** | **12.5** | **100%** |
| Cartpole (DITL-dReal) | >2 (hours) | N/A | N/A | 0% |
| **Cartpole (DITL, main paper)** | **0.9 ± 0.3 (hours)** | **0.021 ± 0.012** | **0.045** | **100%** |
| PVTOL (DITL-dReal) | >24 (hours) | N/A | N/A | 0% |
| **PVTOL (DITL, main paper)** | **13 ± 6 (hours)** | **0.011 ± 0.008** | **0.028** | **100%** |

