# OpenReview forum: "Neural Lyapunov Control for Discrete-Time Systems"
_NeurIPS.cc/2023/Conference — NeurIPS 2023 poster_

### Official Review · Reviewer_KEba · 2023-07-04

**Soundness:** 3 good
**Presentation:** 3 good
**Contribution:** 2 fair
**Rating:** 6
**Confidence:** 5

**Summary:**

The paper proposes a Lyapunov control method for discrete-time systems, in contrast to previous approaches targeted at continuous-time systems. The paper proposes a mixed-integer linear programming approach for verifying stability conditions in discrete-time systems, a technique for computing sub-level sets which define the region of attraction, and a heuristic gradient-based approach for finding counterexamples that can accelerate the learning of Lyapunov functions.

**Strengths:**

I appreciate the counterexample generation method proposed in this work. It leverages the key ideas in adversarial training in the computer vision domain to use gradients to accelerate the discovery of counterexamples.

**Weaknesses:**

- The Heuristic Counterexample Generation cannot guarantee all counterexamples are found. It relies on sampling in the state space, but sampling, the method itself, cannot exhaust the state space. Therefore, the verification is more like an empirical evaluation. I am not saying being “empirical” is not good, but it just simply does not match the verification objective which should be very strict.

- The authors did not explain if it is truly challenging to train in discrete time system compared to training in continuos time system. In fact, the continuous time system and discrete time system can share the same framework in training. See https://arxiv.org/pdf/2101.05436.pdf how they computed dot{h} in the paragraph below (7).

- I would like to discuss with the authors if verification is truly necessary in Learning for Control. As we know, real-world dynamics are much more complex than the dynamics in the neural Lyapunov studies. The verification methods presented in these papers are very difficult to be applied to the real world and make real impacts. Often, a method that pursues excellent empirical performance instead of 100% rigorous verification will be closer to real-world scenarios in control (other domains such as planning or chip design might be different).


**Questions:**

See weaknesses.

**Limitations:**

See weaknesses.

---

> ### Author Rebuttal · Authors · 2023-08-10
>
> We thank the reviewer for the thoughtful and engaging comments. Below we provide our responses.
>
> 1. ***Comment:*** *The Heuristic Counterexample Generation cannot guarantee all counterexamples are found. It relies on sampling in the state space, but sampling, the method itself, cannot exhaust the state space. Therefore, the verification is more like an empirical evaluation.*
>
>     ***Response:*** This appears to be a misunderstanding. In fact, our approach uses both heuristic and sound counterexample generation (using MILP). It is this combination that enables both improved efficacy, as well as soundness of the overall approach in that in the end the Lyapunov conditions are provably satisfied.  In particular, the controller that our algorithm returns is always verified to be stable using the sound MILP verifier that we describe, and it is the stopping criterion of our algorithm.  Consequently, as we assert in Theorem 4.1, our algorithm always returns provable stable control (in the sense of satisfying Lyapunov stability conditions) by construction.
>
>      $\ $
>
> 2. ***Comment:*** *The authors did not explain if it is truly challenging to train in discrete time system compared to training in continuos time system.*
>
>     ***Response:*** As we emphasize in our Responses to Reviewer qMzc above, we can exploit structure in discrete-time settings that cannot be so easily exploited in continuous-time domains. As a result, we actually attain order-of-magnitude improvement compared to prior art, which is mostly in continuous time. Since neural network controllers take non-negligible computation to output control, systems using these are effectively discrete time, and the structure we exploit is therefore highly relevant.
>
>      $\ $
>
> 3. ***Comment:*** *In fact, the continuous time system and discrete time system can share the same framework in training. See https://arxiv.org/pdf/2101.05436.pdf how they computed dot{h} in the paragraph below (7).*
>
>     ***Response:*** Note that in the referenced paper the authors are not doing strict verification, and their evaluation is entirely empirical. Thus, Qin et al. (the referenced paper) approximate the continuous time derivative $\dot{h}$ numerically and still achieve good empirical results.  In contrast, we achieve fully verified stability (see our response to Comment 1 above). Since the definition of continuous and discrete-time Lyapunov stability condition are different, we cannot  use the numerical method for derivatives in the paper, as doing so will not allow the controller-Lyapunov function pair to pass our MILP verifier. More precisely, in continuous-time settings, the key condition is $\forall x \in D; \nabla_{f_u} V(x) \leq 0$, where $\nabla_{f_u} V(x)$ is Lie derivative of function $V$ w.r.t. vector field $f_u$ at point $x$.  In discrete-time settings, the condition is: $\forall x \in D; V(f(x)) - V(x) \leq 0$. Satisfying the former condition does not, in general, satisfy the latter (because discretized continuous-time systems imply that control is effectively piecewise constant, rather than continuously adapting to state feedback).
>
>     $\ $
>
>     In addition the nature of our training framework is itself quite different from Qin et al. (and more akin to the NLC framework) in that we leverage a form of counterexample-guided abstraction and refinement (CEGAR) paradigm, iteratively generating counterexamples and adding these to training data. Our key advances within this framework involve the use of gradient-based heuristic counterexample generation and MILP-based verification, which in combination achieve significant improvements over prior art.
>
>      $\ $
>
>     A final distinction is that Qin et al. are primarily concerned with safety (avoiding unsafe sets), while we focus on Lyapunov stability (ensuring that the system converges to the origin).
>
>      $\ $
>
> 4. ***Comment:*** *I would like to discuss with the authors if verification is truly necessary in Learning for Control. As we know, real-world dynamics are much more complex than the dynamics in the neural Lyapunov studies. The verification methods presented in these papers are very difficult to be applied to the real world and make real impacts. Often, a method that pursues excellent empirical performance instead of 100\% rigorous verification will be closer to real-world scenarios in control (other domains such as planning or chip design might be different).*
>
>     ***Response:*** This is wonderful discussion to have! Indeed, the current state of the art of approaches that yield fully verified properties such as stability cannot be applied to the full complexity of real control problems. Nevertheless, just because we cannot do this now does not mean that we will never have this ability. We believe that our work serves as an important stepping stone in the line of research that continues to improve scalability. It can often takes several years of concerted effort to achieve a high-impact goal, and we feel that the aspiration to scalable synthesis of fully-verified control in real systems is a worthy goal to aspire to. In the meantime, we must indeed engage in a combination of empirical evaluation and testing with verification of smaller subcomponents, but the goal is to be able to verify larger and larger system components over time, until we can verify the full system.

---

> > ### Comment · Reviewer_KEba · 2023-08-10
> > **Thanks for the Response**
> >
> > I appreciate the authors' response to my questions. Since the first weakness is a misunderstanding, I don't have further question on the techinical validity of the proposed approach. I will change to accept.

---

> > > ### Author Response · Authors · 2023-08-11
> > >
> > > We appreciate the reviewer's thoughtful consideration of our paper and responses, and glad that we were able to clarify the misunderstanding.  Please let us know if you have any further questions.

---

### Official Review · Reviewer_ruNW · 2023-07-06

**Soundness:** 3 good
**Presentation:** 3 good
**Contribution:** 2 fair
**Rating:** 5
**Confidence:** 3

**Summary:**

The stability of nonlinear systems has long been a challenge in the field of control systems, with current methodologies employing Lyapunov stability theory to derive control policies. However, finding suitable Lyapunov functions for these systems is notably complex. To address this, researchers have started using neural networks to approximate Lyapunov functions, though primarily for continuous-time systems. The paper introduces a first-of-its-kind approach for learning neural Lyapunov control in discrete-time systems. This approach has three critical components:A unique mixed-integer linear programming method to verify stability conditions.A new method for computing sub-level sets to identify the region of attraction.A heuristic gradient-based method to rapidly find counterexamples, which improves the learning process of Lyapunov functions.Experimental results show a significant improvement over current techniques, outperforming neural Lyapunov control baselines by an order of magnitude in both running time and the size of the region of attraction. For the 'cartpole' and 'PVTOL' benchmarks, this is the first automated method to yield a provably stable controller.

**Strengths:**

This method outperform recent neural Lyapunov control baselines by an order of magnitude in both running time and the size of the region of attraction.

**Weaknesses:**

The loss function is then a weighted sum of several terms, which is hard to balance.

**Questions:**

How does your method compare in terms of advantages to similar works like "Reinforcement Learning Control of Constrained Dynamic Systems with Uniformly Ultimate Boundedness Stability Guarantee" by Han M, Tian Y, Zhang L, et al. (Automatica, 2021, 129: 109689)?

**Limitations:**

This method leverages a mixed-integer linear programming approach, which is known to be NP- hard.

---

> ### Author Rebuttal · Authors · 2023-08-10
>
> We thank the reviewer for the comments. Our detailed responses are below.
>
> 1. ***Comment:*** *The loss function is then a weighted sum of several terms, which is hard to balance.*
>
>     ***Response:*** It is common for loss function to be the weighted sum of several terms, as those weights are hyperparameters for learning we need to calibrate. For example, in both of our baselines, NLC and UNL, their loss function are the weighted sum of several terms. Similarly, standard RL approaches such as PPO also involve many hyperparameters.
>
>     $\ $
>
> 2. ***Comment:*** *How does your method compare in terms of advantages to similar works like "Reinforcement Learning Control of Constrained Dynamic Systems with Uniformly Ultimate Boundedness Stability Guarantee" by Han M, Tian Y, Zhang L, et al. (Automatica, 2021, 129: 109689)?*
>
>     ***Response:*** We would like to clarify that the referenced paper is addressing a fundamentally different problem from ours. Although they also refer to a notion of stability, it is quite distinct, and is more useful for providing safety than stability guarantees. For example, in our paper we want the cartpole to converge to and remain at the origin, while in their experiment they want the carpole position to remain within a target safe interval ($x<4$).
>
>     $\ $
>
>     Due to the formal distinction, the goal of Han et al. is to achieve what is commonly referred to as safety, rather than stability in the Lyapunov sense. We would also like to point out that  Han et al. does not provide any strict verification guarantees like ours (for example, their Theorem 1 only guarantees safety approximately), which makes their evaluation essentially empirical, while our approach returns a controller that is provably stable. Moreover, while Han et al. also utilize Lyapunov functions, the associated conditions are in expectation, rather than uniform over all states, as required in the notion of stability that we use.
>
>     $\ $
>
>     To be more precise, in Han et al. the key concept is uniformly ultimate boundedness (UUB), which is defined as follows (Defn. 1 on page 3; also see Thowsen, 1983): A system is said to be uniformly ultimately bounded (UUB) with ultimate bound $\eta$, if there exist positive constants $b$, $\eta$, where $\forall \epsilon < b$ there exists $T(\epsilon, \eta)$, such that $\|x_{0}\| < \epsilon$ $\Rightarrow$ $\|x_t\| < \eta$, $\forall t > T(\epsilon, \eta)$. At the high level, this means that trajectories that start from inside the safe set (defined by $\|x_{0}\| < \epsilon$) will remain inside a subset of the safe set (defined by $\|x_t\| < \eta$), for suitably chosen $b$ and $\eta$.
>
>     $\ $
>
>     In contrast, the stability notion we are concerned with is defined as follows: a system is stable within a region of attraction $\mathcal{R}$ (which includes the origin) if $\forall x_0 \in \mathcal{R}$, $\lim_{t\rightarrow\infty} x_t = 0$, where $x_{t+1} = f(x_t)$ (we omit the control input here to simplify discussion). At the high level, we require convergence to the origin from any starting point in the region of attraction, which we also aim to make as large as possible.
>
>     $\ $
>
> 3. ***Comment:*** *This method leverages a mixed-integer linear programming approach, which is known to be NP-hard.*
>
>     ***Response:*** Tight verification is typically NP-hard, so that the main avenues for advances in scalability are either in the direction of relaxation (which loses tightness) or taking advantage of problem structure. MILP is an approach that takes advantage of any linearity structre in the problem, and MILP solvers can indeed scale well, although they, too, have their limits. While clearly there is much research that remains in further scaling the learning of provably stable control, we note that our approach is an order-of-magnitude advance in efficacy/scalability over prior art for the more challenging problems.

---

> > ### Author Response · Authors · 2023-08-15
> >
> > Dear Reviewer ruNW.  Thank you again for your review.  As the author-reviewer discussion phase is closing relatively soon, we hope you will be able to take a look at our detailed responses, which address all three of your questions and concerns.  In particular, we would be very happy to field any additional questions that you may have.

---

> > > ### Comment · Reviewer_ruNW · 2023-08-16
> > >
> > > Thanks! I have raised my rating.

---

### Official Review · Reviewer_yt77 · 2023-07-09

**Soundness:** 1 poor
**Presentation:** 2 fair
**Contribution:** 1 poor
**Rating:** 2
**Confidence:** 5

**Summary:**

This paper extends the neural Lyapunov control method to the discrete-time dynamical systems and propose two tricks to quickly find the counterexamples to accelerate the training process. The proposed method is tested in several standard tasks with comparison to the SOTA methods.

**Strengths:**

[1] The proposed mixed-integer LP and PGD method can efficiently find the counterexamples in the training process.

[2] The proposed method can find the largest ROA compared to the existing methods.




**Weaknesses:**

[1] The extension of neural Lyapunov control from continuous-time dynamical systems to discrete-time dynamical systems has little contribution to the existing neural control community because the Lyapunov theories for continuous/discrete system are essentially the same.

[2] The current method has huge computational cost for the case where the high-dimensional tasks are taken into account.

[3] Lemma 3.1 is NOT mathematically correct.





**Questions:**

[1] Does the proposed method scale to the high-dimensional systems such as controlling the molecular dynamics?

[2] The neural Lyapunov method for unknown system (UNL) focuses on the realistic model-free settings, while the proposed method relies on the fully known dynamics, is the comparison in terms of size of ROA fair?

[3] The problematic Lemma 3.1 cannot guarantee the soundness of the proposed method.

**Limitations:**

The authors only test the proposed method in toy models; however, they do not show the current method can be applicable in the real-world scenarios.

---

> ### Author Rebuttal · Authors · 2023-08-10
>
> We thank the reviewer for the thoughtful comments. We respond to these below.
>
> 1. ***Comment:*** *The extension of neural Lyapunov control from continuous-time dynamical systems to discrete-time dynamical systems has little contribution to the existing neural control community because the Lyapunov theories for continuous/discrete system are essentially the same.*
>
>     ***Response:*** As we highlight in our response to Reviewer qMzc (in particular, our response to Comment 1), there is a very consequential difference in the Lyapunov conditions between continuous-time and discrete-time systems. It is this difference that enables us to exploit the structure of the condition in discrete-time systems to enable a substantial increase in efficacy that we exhibit in the experiments.  We also wish to emphasize that stability for continuous-time systems does not imply stability in discrete-time systems, nor is the converse true. For example, in our inverted pendulum experiment, all the the continuous-time controller and Lyapunov function generated by the NLC method fails the discrete-time Lyapunov condition. Moreover, as neural network control takes non-negligible computational time in a real robotic system, the controller is effectively discrete time, and stability guarantees for continuous-time systems would in general be unsound in such settings.
>
> $\ $
>
> 2. ***Comment:*** *The current method has huge computational cost for the case where the high-dimensional tasks are taken into account. Does the proposed method scale to the high-dimensional systems such as controlling the molecular dynamics?*
>
>     ***Response:*** Today, no method exists that can work on systems of such scale with general non-linear dynamics and provable stability guarantees. As our experiments demonstrate, our approach is an order-of-magnitude improvement over the state of the art in higher-dimensional settings, but clearly there is much research that remains to further improve scalability. The key bottleneck in this setting remains sound verification of stability.
>
> $\ $
>
> 3. ***Comment:*** *The neural Lyapunov method for unknown system (UNL) focuses on the realistic model-free settings, while the proposed method relies on the fully known dynamics, is the comparison in terms of size of ROA fair?*
>
>     ***Response:*** We appreciate this point. We wish to clarify that the comparison is not intended to make claims about efficacy of UNL, which is indeed targeted at a far more challenging problem; UNL simply serves as another state of the art baseline. Not including it would have exposed us to a criticism that UNL is a more recent approach and could be a more competitive baseline than others (which, as we show, is not the case).
>
> $\ $
>
> 4. ***Comment:*** *The authors only test the proposed method in toy models; however, they do not show the current method can be applicable in the real-world scenarios.*
>
>     ***Response:*** We observe that both of our baseline papers, UNL and NLC, utilize such control dynamics. Indeed, the benchmarks we use are common within the control community. Our aim is for our work to serve as a stepping stone towards practical application in real-world scenarios.

---

> > ### Comment · Reviewer_yt77 · 2023-08-12
> >
> > Thank you very much for the response.    I will leave the scores as is.

---

> > > ### Author Response · Authors · 2023-08-13
> > >
> > > Thank you for reading our response.  Would you be able to clarify why you did not find our responses convincing?  We had addressed all of the weaknesses, questions, and limitations that the reviewer had raised.  Perhaps the most compelling evidence that our paper significantly advances the state of the art comes from our experimental evidence, in comparison with prior approaches.  We explain, both in the paper, and in responses to other reviewers, what technical advances enabled the considerable improvements we observe in the experiments.  In particular, what may perhaps be a misunderstanding is that our key observation is that Lyapunov conditions for discrete-time systems possess important structure that we take advantage of.

---

> > > > ### Comment · Reviewer_yt77 · 2023-08-17
> > > >
> > > > Mathematically, this reviewer still believes that the essence of the current framework is the same as that for the continuous-time model.   I don't agree with the response by the authors for the difference between the discrete- and the continuous-time dynamical systems.  Superficially, the stability criteria are different for two different systems; however, the essences are the same, not bringing novel stuffs.  The results could be publishable; however, the NeurIPS, as a top-tier platform, is not an appropriate venue for this case.   I am really sorry that I cannot be more positive from my experience for this case.

---

> > > > > ### Author Response · Authors · 2023-08-17
> > > > >
> > > > > We appreciate the reviewer's continued engagement. Perhaps it is useful to step back for a moment. We surely agree that the two notions *are* different; we are not inclined to litigate here whether this difference is large or small, but simply observe that it exists. Now, our goal is a system with *provable stability guarantees*. While empirically, small differences are often inconsequential, when you aim to achieve provable guarantees, the smallest and most superficial seeming differences can prove consequential. Therefore, we argue that the only relevant question is whether the difference in the Lyapunov condition between continuous-time and discrete-time systems is consequential in regards to the goal of effectively learning provably Lyapunov stable control. Our paper demonstrates that it is. We now explain why in some detail.
> > > > >
> > > > > Recall that in continuous-time settings, the key condition is $\forall x \in D; \nabla_{f_u} V(x) \leq 0$, where $\nabla_{f_u} V(x)$ is Lie derivative of function $V$ w.r.t. vector field $f_u$ at point $x$.  In discrete-time settings, it is: $\forall x \in D; V(f(x)) - V(x) \leq 0$. If $V$ is a neural network with ReLU activations, it is not continuously differentiable, which makes it non-obvious how we can construct a MILP to verify the condition on the Lie derivative uniformly over the state space that is needed in continuous-time systems. However, this issue does not come up for the Lyapunov condition in discrete-time systems. An alternative could be to replace ReLU units with continuously differentiable neurons, but these must now be approximated in an MILP formulation using piecewise linear functions, which is a major challenge to scalability and efficacy, as it introduces an approximation gap that makes the Lyapunov condition even more difficult to achieve as we get closer to the origin. Another alternative is to use dReal for verification, but as we demonstrate above (see the data in response to Reviewer NZiv, Comment 2), MILP is in fact a critical ingredient in enabling high efficacy of the proposed approach.
> > > > >
> > > > > The reviewer's intuition that the differences are insignificant is, indeed, an argument for the significance of our contribution. Superficially, the differences may appear small. However, our experiments demonstrate that they are, in fact, highly consequential in that we can fruitfully exploit the particular structure in the discrete-time Lyapunov condition which is absent in its continuous-time counterpart. In particular, the combination of (a) heuristic verification techniques that we propose, and (b) MILP-based verifier that exploits special structure in the discrete-time Lyapunov condition, yields a dramatic improvement in efficacy, as demonstrated by the fact that on two of the more complex domains in our experimental study, our approach *is the only one that successfully learns a provably stable controller*. Moreover, as our ablation studies demonstrate, both of these contributions are critical to the results.

---

> > > > > > ### Comment · Reviewer_yt77 · 2023-08-19
> > > > > >
> > > > > > Thank you very much for your reply to my previous comments.
> > > > > >
> > > > > > In the previous review and comments, I have emphasized the similarity and the difference between the stability results on continuous- and discrete-time dynamical systems.  Definitely, they have the difference; however, the essences based for deriving stability criteria are the same.  Probably, these authors may not know there is a uniform math framework, dynamical systems on time-scale, for different time-type of dynamical systems.  All the stability criteria could be include in one framework.   The so-called Lie derivative and the difference computations along the considered systems are essentially the same; all are called the Lyapunov second method or direct method.  Actually, even for the continuous-time system, we allow the discussion for the discontinuity in V function or/and in the vector field.   I have mentioned that the current results if correct could be publishable. Yet, NeurIPS is NOT an appropriate venue since the presented contribution does not seem to be that significant.
> > > > > >
> > > > > > Now, after a more care check, I have to report a fatal mistake emergent in the current work.  The Lemma 3.1 is NOT mathematically correct.  Since the difference between V(f(x))-V(x) upper bounded by only a negative Constant definitely leads to a contradiction and indeed an empty set where the conditions in this Lemma are valid there.  Therefore, I cannot imagine how the convergence could be numerically made by the current incorrect theory and the correspondingly developed algorithm.   There are a huge amount of rigorous results in the control and cybernetics literature on infinite or finite-time stability for continuous- and discrete-time dynamical systems.   The mistake that was found in this manuscript will certainly leads me to the recommendation against the acceptance of this work.     I also suggest the other experts to pay their attentions on this problem.
> > > > > >
> > > > > > I still believe that the essence driven by the rigorous mathematical foundation is crucial for a novel work, especially in the recent area of AI.  I am sorry that I still cannot be more positive on this work.  Due to the fatal mistake, I therefore will consider to decrease the score.

---

> > > > > > > ### Author Response · Authors · 2023-08-21
> > > > > > >
> > > > > > > We appreciate the reviewer's continued engagement and comments.
> > > > > > >
> > > > > > > **Regarding Lemma 3.1.** There appears to be a misunderstanding, and we sincerely apologize for not being clear when stating Lemma 3.1.  In our footnote 1 on page 3 (prior to the lemma) we mentioned that we introduce a constraint $|| x||\geq \epsilon$ (all norms are $\infty$-norm), where $\epsilon$ is a small pre-defined parameter, in order to mitigate numerical instability of verification. This constraint does not affect the overall convergence and is also employed in the baseline papers [Chang et al., 2019, Zhou et al., 2022]. Throughout the paper, this constraint is treated as a prior. We establish the Lyapunov condition for $|| x|| \geq \epsilon$ only, and our controller drives the system to converge to $|| x||< \eta$, a small region close to the origin. In Lemma 3.1, the symbol $\zeta$ should actually be $\zeta (\epsilon)$, indicating that the choice of the negative constant $\zeta$ depends on the chosen value of $\epsilon$ in order to ensure that the condition in Lemma 3.1 does not induce an empty set. Consequently, it will not lead to any contradiction when the conditions in Lemma 3.1 are met. Our experiments demonstrate this.
> > > > > > >
> > > > > > > We acknowledge that the footnote could easily be overlooked by the reader, and the proof sketch is, of course, incomplete (we did not have the space to include all of the details, and prioritized the algorithmic and empirical aspects of the problem). Nevertheless, the reviewer's point is well taken, and we will add a more precise statement and proof in the revision that also shows that we are both guaranteed to reach the $\eta$-ball around the origin in finite time, and remain in this ball thereafter. Specifically, we will revise Lemma 3.1 and its complete proof as follows:
> > > > > > >
> > > > > > >
> > > > > > > **Lemma 3.1:** Let $L_f$ and $L_v$ denote the Lipschitz coefficients of $\overline{f}(x)=f(x,\pi_\beta(x))$ and $V_\theta$, respectively. Suppose there exist positive constants $\epsilon$,  $\rho$, $\gamma$, and a negative constant $\zeta$ such that the following properties hold: (a) $V_{\theta}(f(x,\pi_\beta(x))) - V_{\theta}(x) < \zeta$ for all $x$ with $\epsilon \leq ||x|| \leq \gamma$, (b) {$x: V_\theta(x) \leq \rho$}$ \subseteq B(0,\gamma)$ (where the ball is with respect to the $\infty$-norm), (c) $V_\theta(0) = 0$,  (d) $f(0,\pi_\beta(0)) = 0$, and (e) $L_{v}L_{f}\epsilon \leq \rho$. Then the following properties hold: (i) The set {$x: V_\theta(x) \leq \rho$} is positive invariant under dynamics $f$ and control policy $\pi_\beta$, and (ii) If $V_\theta(x_0) \leq \rho$, then there exists $K \in ${$0,\ldots,\lceil \frac{\rho}{-\zeta} \rceil $} such that $||x_{K}|| < \epsilon$, and $||x_k|| < \eta$ for all $k > K$, where $\eta$ is the radius of the smallest $\infty$-norm ball containing the set {$x: V_{\theta}(x) \leq L_{v}L_{f}\epsilon$}.
> > > > > > >
> > > > > > > **Proof:** We first prove property (i). Suppose that $V_\theta(x) \leq \rho$. If $||x|| \in [\epsilon,\gamma]$, then $V_\theta(f(x,\pi_\beta(x))) < V_\theta(x) \leq \rho$ by conditions (a) and (b). If $||x|| \leq \epsilon$, then $V_\theta(f(x,\pi_\beta(x))) = |V_\theta(f(x,\pi_\beta(x)))-V_\theta(f(0,\pi_\beta(0)))|$ by conditions (c) and (d), and thus $V_\theta(f(x,\pi_\beta(x))) \leq L_{f}L_{v}||x|| \leq \rho$ by condition (e). Hence property (i) holds.
> > > > > > >
> > > > > > > To prove property (ii), suppose that $V_\theta(x_0) \leq \rho$. Since {$x: V(x) \leq \rho$} is positive invariant by the above, we must have that $||x_k|| \leq \gamma$ for all $k \geq 0$ by condition (b). Suppose that, for all $k^{\prime} \in ${$0,\ldots \lceil \frac{V(x_0)}{-\zeta} \rceil $}, we have $||x_{k^{\prime}}|| > \epsilon$. Then at time $k^{\prime} = \lceil \frac{V(x_0)}{-\zeta} \rceil$, we have that $V(x_{k^{\prime}}) = V(x_0) + \zeta k^{\prime} < 0$, contradicting positive-definiteness of $V$. This contradiction, together with the fact that $V_\theta(x_0) \leq \rho$, implies that $||x_K|| \leq \rho$ as in property (ii). Finally, by the discussion of the preceding paragraph, the set {$x: V_{\theta}(x) \leq L_{f}L_{v}\epsilon$} is positive invariant and contained in $B(0,\eta)$, and $||x_k||< \epsilon$ implies $V_{\theta}(x_{k+1}) \leq L_{f}L_{v}\epsilon$. Hence $||x_k|| \leq \eta$ for all $k > K$. This completes the proof.
> > > > > > >
> > > > > > > We observe that, within the network structures considered in our experiments, $\eta$ can be made arbitrarily small by reducing $\epsilon$, i.e., we can prove that the state converges to and remains within an arbitrarily small ball centered at the origin.
> > > > > > >
> > > > > > >
> > > > > > > We confirmed that all the conditions in Lemma 3.1 held in our numerical experiments. The negative constant $\zeta$ choosen in our experiments was -0.00001, which ensured that the conditions in Lemma 3.1 are meaningful. In any case, connecting all the relevant pieces here amounts to exploiting the Lipschitz continuity assumption on $f$, and Lipschitz continuity of neural networks with ReLU activations.

---

> > > > > > > > ### Comment · Reviewer_yt77 · 2023-08-21
> > > > > > > >
> > > > > > > > This reviewer thanks the efforts made by the authors of this work.
> > > > > > > >
> > > > > > > > However, clearly, the footnote 1 was only used for avoiding the numerical instability in simulations, not for realizing the stabilization of the equilibrium rigorously.   Once Lemma 3.1 is fixed as the manner proposed in the above reply, the convergence of the equilibrium is no longer ensured mathematically and rigorously.   Without a solid foundation, all the stabilization results obtained numerically in the present work is NOT convincing yet.   Actually, there were compact and correct results from theoretical viewpoint to resolve the fatal problem in the literature even ten years ago.  I believe with these stability results for discrete-time systems, the algorithm could be amended into a correct manner.   Unfortunately, control protocol proposed in the reply is not correct, not assuring the stability of the equilibrium, but at most the attracting property of the small neighborhood of the equilibrium.
> > > > > > > >
> > > > > > > > Even using the proposed Lemma 3.1 for designing the numerical application, one needs to make a lot of discussions on the choice of the parameters including \epsilon.  This has not been completely done yet in the current work, which could be included in the future work.  Clearly, the core problem of the stabilization cannot be resolved at the present stage, which leads me definitely to sustaining my recently amended score.

---

> > > > > > > ### Author Response · Authors · 2023-08-21
> > > > > > >
> > > > > > > **Regarding Dynamical Systems on time-scale.** The unified framework of dynamic Lyapunov equations on time scales is certainly elegant, but orthogonal to our goals.  The former is a mathematical unification; our interest, however, is to develop new computational methods for checking stability of a discrete-time system. We are not trying to derive new conditions for discrete-time stability. The reviewer is correct that the discrete and continuous time Lyapunov direct methods are very similar; however, note that the algorithms for stability checking can be different. This difference allows us to exploit specific structures in the Lyapunov conditions in discrete-time systems and thereby obtain a learning-based approach with significantly better performance that all state-of-the-art alternatives.
> > > > > > >
> > > > > > > At the high level, we feel that our perspective on the problem is different from that of the reviewer.  The reviewer views it predominantly through a mathematical lens. On the other hand, our perspective is largely computational and empirical. We hope that the reviewer can consider this contribution from an alterantive perspective of developing a learning algorithm that exploits problem structure to computational advantage, and yields compelling empirical results. We argue that contributions of this kind are no less significant in AI than work focused primarily on foundational mathematical issues.

---

> > > > > > > > ### Comment · Reviewer_yt77 · 2023-08-21
> > > > > > > >
> > > > > > > > Mathematical formulation of a problem is so important.   If the authors knew very much about the stability theories for different time-type of dynamical systems, the current mistake would not happen.   I am not argue with you the importance of the mathematics, but my point is: A correct mathematical principle can assure the correct development direction of engineering work.   If the theory is wrong or not complete (this work unfortunately falls into this case), then how to guarantee the stability as expected in real applications.   All results are empirical, which could be publishable, not in this venue of NeurIPS.
> > > > > > > >
> > > > > > > > I am really sorry that I cannot be more positive.

---

> > > > > > > > > ### Author Response · Authors · 2023-08-21
> > > > > > > > >
> > > > > > > > > We appreciate the reviewer's quick response.  In a nutshell, the two main issues raised by the reviewer are summarized below, with our comments about them.
> > > > > > > > >
> > > > > > > > > ***Comment:*** *Unfortunately, control protocol proposed in the reply is not correct, not assuring the stability of the equilibrium, but at most the attracting property of the small neighborhood of the equilibrium.*
> > > > > > > > >
> > > > > > > > > ***Response:*** We regret the reviewer's use of hyperbolic language of "incorrect" here.  Because of numerical precision limits, in general all computational methods can ensure at most the attracting property of the small neighborhood of the equilibrium. We thought this was clear given prior work (such as [Chang et al., 2019, Zhou et al., 2022]) that we are after a weaker form of stability than conventional, but can see that the reviewer objects to calling it "stability".  This is a criticism of this entire line of work, and one we disagree with.  In any case, this issue is not a mistake, but a difference in the meaning of a term (that is, the term is indeed effectively overloaded in the literature on learning stable controllers). What is clear is that empirically this distinction is quite inconsequential (that is, reaching close enough to the origin suffices).
> > > > > > > > >
> > > > > > > > > $\ $
> > > > > > > > >
> > > > > > > > > ***Comment:*** *All results are empirical, which could be publishable, not in this venue of NeurIPS.*
> > > > > > > > >
> > > > > > > > > ***Response:*** Correct, this is a computational and empirical work.  We do not claim to develop new theory, but simply leverage known theory of Lyapunov stability in discrete-time dynamical systems. It is worth mentioning that two of our main baselines [Chang et al., 2019, Zhou et al., 2022], were both published at NeurIPS.
> > > > > > > > >
> > > > > > > > > $\ $
> > > > > > > > >
> > > > > > > > >
> > > > > > > > > At this point, it is clear that the reviewer and the authors are effectively speaking different languages, with the same words having a different meaning.  It is likely that we come from somewhat different subcommunities of NeurIPS with different publishing styles and traditions.

---

> > > > > > > > > > ### Comment · Reviewer_yt77 · 2023-08-21
> > > > > > > > > >
> > > > > > > > > > 1. Clearly, in the manuscript, the authors claimed the efficacy of their neural control protocol in asymptotical stabilization of the equilibrium.  The definition of this stability was clearly stated from lines 95 to 96 in the original manuscript, requiring x(t) approaching zero, NOT wandering in the neighborhood of the equilibrium (as defined in the reply).   The revised Lemma 3.1 can only assure the boundedness of the trajectory, but cannot guarantee the asymptotical stability of the equilibrium.   This is the typical and scientific language in the communities of control and cybernetics.  If one leverages the theory of control and machine learning techniques to solve typical control problems, the elementary and standard terminology should be taken into account.
> > > > > > > > > >
> > > > > > > > > > More importantly, the authors clearly stated their definition of stability in their response to the comment #2 from Reviewer ruNW.
> > > > > > > > > >
> > > > > > > > > > "In contrast, the stability notion we are concerned with is defined as follows: a system is stable within a region of attraction  (which includes the origin) if..."
> > > > > > > > > >
> > > > > > > > > > If they are going to change their definition, then the entire work, as well as the basic mission, should be completely and substantially rewritten.
> > > > > > > > > >
> > > > > > > > > > 2. To be frank, the mistake actually could be amended; however, the authors did not make a good of the hint given by this reviewer since there are CORRECT and existing theories to guarantee the stability of the equilibrium for discrete-time dynamical systems in control literature, not in math literature.
> > > > > > > > > >
> > > > > > > > > > 3. Probably, we do not use the same system of language because this reviewer not only focuses on the rigorous theory but also the empirical and feasible control methods.
> > > > > > > > > >
> > > > > > > > > > 4. Actually, all the continuous-time system, when put into simulations, should be discretized as a discrete-time system; otherwise the computation is not feasible.  Understanding the basic principles of different time-scale of dynamical systems could be beneficial to avoiding a trap of common mistakes emergent in control community.
> > > > > > > > > >
> > > > > > > > > > 5. Here, I do not intend to comment the previous works mentioned by the authors.  Indeed, I do not agree with the argument that they regarded as this work to be in the direction the same as the previous ones.   Actually, the previous works presented the correct theory, allowing the minor errors in simulations because of the discretization.   However, these authors actually go in an opposite way: using the incorrect or incomplete stability theory (see Lemma 3.1, their definition (lines 95-96) in original manuscript, and their reply to the other Reviewer ruNW's comment #2), generating the simulations whose fidelity are not thoroughly investigated.
> > > > > > > > > >
> > > > > > > > > > Above all, this work now is deviating from the original intention and needs substantial rewritten.   Minor revision is not applicable.

---

### Official Review · Reviewer_qMzc · 2023-07-09

**Soundness:** 3 good
**Presentation:** 3 good
**Contribution:** 3 good
**Rating:** 6
**Confidence:** 2

**Summary:**

This paper studies learning stabilizing controllers with neural Lyapunov functions in discrete-time nonlinear systems. Previous works on neural Lyapunov control focused on continuous-time systems, and this is the first work that studies discrete-time systems. The outline of the proposed approach is similar with prior works on continuous-time systems, while the authors propose several key novel techniques for each step: They only use ReLU activations in the neural Lyapunov functions so that the constraints can be verified by mixed-integer programming. The authors also proposed a heuristic way to generate counterexamples efficiently via projected gradient descent. The simulation results show that the proposed method can find larger ROA and be implemented efficiently in several examples.

**Strengths:**

Learning neural Lyapunov control in discrete-time systems is important for many applications. This work proposes many novel techniques to handle the difficulties that have been found in prior works. The simulation results are promising.

**Weaknesses:**

I have some minor concerns about the presentation. After reading this paper, it is not clear to me which challenges are specific to the discrete-time system.

In the first part of verifying Lyapunov constraints, the authors take the motivation from SMT solvers. However, this is also a challenge for neural Lyapunov control in continuous-time systems. I hope the authors can add a discussion about what efforts have been made to improve efficiencies in the continuous-time systems and why we cannot directly apply them here. And for the proposed ReLU network plus mixed-integer programming, does this method also apply to continuous-time systems? Or is it limited to discrete-time systems because of some special properties?

There is a similar question for proposed heuristic for generating counterexamples. Is this heuristic limited to discrete-time system? If not, can we replace the counterexample generation step in existing algorithms for continuous-time system and see if the performance improves?

Another minor concern I have with the PGD heuristic is that it might fail to find a counterexample because gradient-based search can get stuck in local minimums. Thus, in the final step when the algorithm claims success, shall we use a less efficient way to verify the Lyapunov conditions are actually satisfied?


**Questions:**

Please see my comments in the previous section.

**Limitations:**

I didn't see the discussion about limitations or future directions in my read. But I don't think there is any potential negative societal impact.

---

> ### Author Rebuttal · Authors · 2023-08-09
>
> We appreciate the reviewer's thoughtful and detailed comments and questions, to which we respond in detail below.
>
> 1. ***Comment:*** *It is not clear to me which challenges are specific to the discrete-time system.*
>
>     ***Response:*** There is a consequential distinction in the definition of Lyapunov functions between continuous and discrete-time settings.  In continuous-time settings, the key condition is $\forall x \in D; \nabla_{f_u} V(x) \leq 0$, where $\nabla_{f_u} V(x)$ is Lie derivative of function $V$ w.r.t. vector field $f_u$ at point $x$.  In discrete-time settings, it is: $\forall x \in D; V(f(x)) - V(x) \leq 0$. If $V$ is a neural network with ReLU activations, it is not continuously differentiable, which makes it non-obvious how we can construct a MILP to verify the condition on the Lie derivative uniformly over the state space that is needed in continuous-time systems. However, this issue does not come up for the Lyapunov condition in discrete-time systems. An alternative could be to replace ReLU units with continuously differentiable neurons, but these must now be approximated in an MILP formulation using piecewise linear functions, which is a major challenge to scalability and efficacy, as it introduces an approximation gap that makes the Lyapunov condition even more difficult to achieve as we get closer to the origin. Another alternative is to use dReal for verification, but as we demonstrate above (see the data in response to Reviewer NZiv, Comment 2), MILP is in fact a critical ingredient in enabling high efficacy of the proposed approach.
>
> $\ $
>
> 2. ***Comment:*** *In the first part of verifying Lyapunov constraints, the authors take the motivation from SMT solvers. However, this is also a challenge for neural Lyapunov control in continuous-time systems. I hope the authors can add a discussion about what efforts have been made to improve efficiencies in the continuous-time systems and why we cannot directly apply them here.*
>
>     ***Response:*** Thank you for this suggestion! We note that our baselines already include state of the art approaches for continuous-time neural Lyapunov control; these still make use of dReal, and no one has proposed effective heuristic techniques for improving verification during training for continuous-time neural Lyapunov control.  Indeed, the conceptual advances we make, such as heuristic verification techniques during training, could in principle be considered also in continuous-time control settings. However, it does not seem evident how to develop MILP-based approaches in that setting (see our response above), and dReal remains a significant bottleneck.
>
> $\ $
>
> 3. ***Comment:*** And for the proposed ReLU network plus mixed-integer programming, does this method also apply to continuous-time systems? Or is it limited to discrete-time systems because of some special properties?
>
>     ***Response:*** As we discuss in our response to Comment 1 above, it is not clear how to adopt our MILP approach to verify Lyapunov conditions in continuous-time systems.
>
> $\ $
>
> 4. ***Comment:*** There is a similar question for proposed heuristic for generating counterexamples. Is this heuristic limited to discrete-time system? If not, can we replace the counterexample generation step in existing algorithms for continuous-time system and see if the performance improves?
>
>     ***Response:*** We believe that the proposed heuristic counterexample generation approach would indeed extend to continuous-time systems. However, we wish to emphasize that the key bottleneck for continuous-time systems remains the scalability of dReal. For instance, even in verifying LQR solutions (which are easy to compute), dReal was unable to complete verification within the specified time frame for both the cartpole (2 hours) and PVTOL (24 hours) control systems. This implies that even if we were to apply the suggested heuristic to continuous-time systems, dReal would still encounter challenges in cases like cartpole and PVTOL (our final step for stopping the algorithm is to pass the verification). It is ultimately the combination of the use of heuristic counterexample generation with MILP-based verification that enables us to make significant progress.
>
> $\ $
>
> 5. ***Comment:*** *Another minor concern I have with the PGD heuristic is that it might fail to find a counterexample because gradient-based search can get stuck in local minimums. Thus, in the final step when the algorithm claims success, shall we use a less efficient way to verify the Lyapunov conditions are actually satisfied?*
>
>     ***Response:*** We would like to clarify that we achieve provable stability precisely by resorting to full MILP-based verification whenver the PGD heuristic can no longer identify counterexamples. Moreover, the MILP-based verifier can generate a counterexample, which then generates additional training iterations (e.g., often PGD now again generates counterexamples once this one is added to the training dataset and can be used to bootstrap the heuristic). Consequently, there are typically several times during training that we have to call the MILP-based verifier. We only claim success when training returns with MILP proving that stability conditions hold. The soundness of our approach is formally asserted in Theorem 4.1, and holds by construction.

---

> > ### Comment · Reviewer_qMzc · 2023-08-17
> >
> > Thank you for the detailed response. I don't have other questions.

---

### Official Review · Reviewer_NZiv · 2023-07-25

**Soundness:** 3 good
**Presentation:** 3 good
**Contribution:** 3 good
**Rating:** 7
**Confidence:** 3

**Summary:**

This paper presents an algorithmic framework for learning neural Lyapunov control with stablizing control policy for discrete-time dynamic systems. It proposes to verify the stability condition with mixed-integer linear programming and speed up Lyapunov function learning with  gradient based approximation. On four benckmarks, it outperforms SOTA baselines in terms of running time, size of regions of attraction and successful rate.

**Strengths:**

- The paper is well structured and nicely presented
- The approach has extended the learning of Lyapunov functions to discrete-time dynamic systems, and has been shown effective on 4 benchmarks with a significant improvement.


**Weaknesses:**

See questions and limitations.

**Questions:**

- The paper provides 2 options for setting the bias term in the training loss function. However, it is not crystal clear the impact of this term over the learning.  An ablation study over the bias setting could help.

- To better motivate the use of mixed-integer linear programming for stability verification in discrete-time systems, it would be good to show a case by comparing DITL-MILP with DITL-dReal.

- Does the run time include the initialization time (for example by LQR)?

**Limitations:**

- The method is built upon the assumption that both the verification function and control policy can be simply represented with MLPs with ReLU activations.

---

> ### Author Rebuttal · Authors · 2023-08-09
>
> We appreciate the reviewer's thoughtful comments and suggestions.  Our detailed responses are below.
>
> 1. ***Comment:*** *The paper provides 2 options for setting the bias term in the training loss function. However, it is not crystal clear the impact of this term over the learning. An ablation study over the bias setting could help.*
>
>     ***Response:*** We appreciate this suggestion.  In our method the bias term is treated as a hyperparameter for learning, thus the one presented in the main paper is the optimal one. Below is the ablation result for using the alternative method (i.e., without bias if the paper uses bias, or with bias when the results in the paper do not):
>
>
> | Environment |  Runtime  |ROA |Max ROA |Success Rate |
> | -------- | -------- | -------- | -------- |-------- |
> | Inverted Pendulum (without bias)         | 11.1 $\pm$ 0.2  (s)   |45 $\pm$ 17     |72 |100\%     |
> | Path Tracking (RL, without bias)         | 17.8 $\pm$ 0.1  (s)   | 8 $\pm$ 0     |8 |100\%     |
> | Path Tracking (LQR, without bias)       | 12.1 $\pm$ 2.9  (s)   |9 $\pm$ 2     |11 |100\%     |
> | Cartpole (with bias)       | >2 (hours)    | N/A  |N/A | 0\%     |
> | PVTOL (with bias)        | 17 $\pm$ 10 (hours)  |   $\sim$ 0 $\pm$ 0 |0.0001 |20\%     |
>
> $\ $
>
> 2. ***Comment***: *To better motivate the use of mixed-integer linear programming for stability verification in discrete-time systems, it would be good to show a case by comparing DITL-MILP with DITL-dReal.*
>
>     ***Response:*** Thank you for the suggestion!  Below are the results for DITL-dReal, in which we use our framework, but replace all the MILP components with dReal.  As we can see, using MILP is an essential ingredient in the success of the proposed approach:
>
> | Environment |   Runtime  |ROA|Max ROA |Success Rate |
> | -------- | -------- | -------- | -------- | -------- |
> | Inverted Pendulum (DITL-dReal)         | **6.0 $\pm$ 1.7** (s)  | 57 $\pm$ 24  |75|**100\%**     |
> | **Inverted Pendulum (DITL, main paper)**        |   8.1 $\pm$ 4.7(s)  |  **61 $\pm$ 31** |**123**|  **100\%**  |
> | Path Tracking (RL, DITL-dReal)        | 600 $\pm$ 0 (s)   | 0 $\pm$ 0  |0 | 0\%     |
> | **Path Tracking (RL, DITL,  main paper)**         | **14 $\pm$ 11** (s)  | **9 ± 3.5** |**16**|   **100\%** |
> | Path Tracking (LQR, DITL-dReal)         | 420.9 $\pm$ 182 (s) | 4 $\pm$ 4   |11 |60\%     |
> | **Path Tracking (LQR, DITL,  main paper)**         | **9.8 $\pm$ 4** (s)  | **8 ± 3**  |**12.5**|  **100\%**  |
> | Cartpole (DITL-dReal)       | >2  (hours)   | N/A  |N/A | 0\%     |
> | **Cartpole (DITL, main paper)**       | **0.9 $\pm$ 0.3**  (hours) | **0.021 $\pm$ 0.012** | **0.045** | **100\%**     |
> | PVTOL (DITL-dReal)       | >24  (hours)   |  N/A   |N/A|0\%     |
> | **PVTOL (DITL, main paper)**       | **13 $\pm$ 6** (hours)  | **0.011 $\pm$ 0.008** | **0.028** | **100\%**     |
>
> $\ $
>
> 3. ***Comment:*** *Does the run time include the initialization time (for example by LQR)?*
>
>     ***Response:*** Run time does not include initialization time as it is negligible in most cases. The run time for LQR for all instances is less than 0.1s. One exception is the run time for RL (path tracking), which is 154.96 seconds as  documented in the results section (Line 330).
>
> $\ $
>
> 4. ***Comment:*** *The method is built upon the assumption that both the verification function and control policy can be simply represented with MLPs with ReLU activations.*
>
>     ***Response:*** This is a good point. Nevertheless, we do not view this assumption as very limiting for our purposes, since NN with ReLU activation layers can approximate arbitrary continuous functions. Furthermore, policies with such structure are common in deep reinforcement learning. In any case, we will comment on this in the revision.

---

> > ### Comment · Reviewer_NZiv · 2023-08-17
> >
> > Thank you for the detailed response. The rebuttal has clearly clarified my questions and I would like to keep my current scores!

---

### Author Response · Authors · 2023-08-21
**Overall Comment to the Reviews**

We thank all reviewers for their comments and valuable feedback. Below we summarize our overall (main) response to the reviewers' questions and suggestions.

1. (reviewer NZiv) We have conducted two sets of additional experiments: 1) ablation study of the bias term; 2) DITL-dReal, where the MILP solver in our algorithm is replaced by the dReal solver. In both cases, our approach achieves the best results.

2. (reviewer qMzc, KEba)  We have explained the challenges that are specific to the discrete-time system versus continuous time, and that we could leverage the specific structure of discrete-time system to significantly improve performance.

3. (reviewer qMzc, KEba) We have clarified that our work provides strict verification. We use both heuristic and sound counterexample generation (using MILP). That is, we achieve provable stability by resorting to full MILP-based verification whenever the PGD heuristic can no longer identify counterexamples.

4. (reviewer ruNW) We have clarified that the paper referenced by the reviewer solves a different problem from ours. Specifically, it is more useful for providing safety than stability guarantees.


5. (reviewer yt77) We argue that computational and empirical work also has significant value in the AI community, and disagree with the reviewer's view that this kind of work is not publishable at NeurIPS. It is worth mentioning that two of our main baselines [Chang et al., 2019, Zhou et al., 2022], were both published at NeurIPS. As of the dynamical systems on time-scale framework mentioned by the reviewer, it is orthogonal to our goals. The former is a mathematical unification; our interest, however, is to develop new computational methods for checking stability of a discrete-time system.


6. (reviewer yt77) We clarified and made more precise the claim in Lemma 3.1, and provided its complete proof. The lemma provides theoretical guarantee that our controller will both guaranteed to reach the $\eta$-ball around the origin in finite time, and remain in this ball thereafter. To clarify,  this $\eta$-ball variation on traditional Lyapunov stability is also used in our baselines [Chang et al., 2019, Zhou et al., 2022], to mitigate numerical precision and stability issues that inevitably arise.

**Lemma 3.1:** Let $L_f$ and $L_v$ denote the Lipschitz coefficients of $\overline{f}(x)=f(x,\pi_\beta(x))$ and $V_\theta$, respectively. Suppose there exist positive constants $\epsilon$, $\rho$, $\gamma$, and a negative constant $\zeta$ such that the following properties hold: (a) $V_{\theta}(f(x,\pi_\beta(x))) - V_{\theta}(x) < \zeta$ for all $x$ with $\epsilon \leq ||x|| \leq \gamma$, (b) {$x: V_\theta(x) \leq \rho$}$ \subseteq B(0,\gamma)$ (where the ball is with respect to the $\infty$-norm), (c) $V_\theta(0) = 0$, (d) $f(0,\pi_\beta(0)) = 0$, and (e) $L_{v}L_{f}\epsilon \leq \rho$. Then the following properties hold: (i) The set {$x: V_\theta(x) \leq \rho$} is positive invariant under dynamics $f$ and control policy $\pi_\beta$, and (ii) If $V_\theta(x_0) \leq \rho$, then there exists $K \in ${$0,\ldots,\lceil \frac{\rho}{-\zeta} \rceil $} such that $||x_{K}|| < \epsilon$, and $||x_k|| < \eta$ for all $k > K$, where $\eta$ is the radius of the smallest $\infty$-norm ball containing the set {$x: V_{\theta}(x) \leq L_{v}L_{f}\epsilon$}.

**Proof:** We first prove property (i). Suppose that $V_\theta(x) \leq \rho$. If $||x|| \in [\epsilon,\gamma]$, then $V_\theta(f(x,\pi_\beta(x))) < V_\theta(x) \leq \rho$ by conditions (a) and (b). If $||x|| \leq \epsilon$, then $V_\theta(f(x,\pi_\beta(x))) = |V_\theta(f(x,\pi_\beta(x)))-V_\theta(f(0,\pi_\beta(0)))|$ by conditions (c) and (d), and thus $V_\theta(f(x,\pi_\beta(x))) \leq L_{f}L_{v}||x|| \leq \rho$ by condition (e). Hence property (i) holds.

To prove property (ii), suppose that $V_\theta(x_0) \leq \rho$. Since {$x: V(x) \leq \rho$} is positive invariant by the above, we must have that $||x_k|| \leq \gamma$ for all $k \geq 0$ by condition (b). Suppose that, for all $k^{\prime} \in ${$0,\ldots \lceil \frac{V(x_0)}{-\zeta} \rceil $}, we have $||x_{k^{\prime}}|| > \epsilon$. Then at time $k^{\prime} = \lceil \frac{V(x_0)}{-\zeta} \rceil$, we have that $V(x_{k^{\prime}}) = V(x_0) + \zeta k^{\prime} < 0$, contradicting positive-definiteness of $V$. This contradiction, together with the fact that $V_\theta(x_0) \leq \rho$, implies that $||x_K|| \leq \rho$ as in property (ii). Finally, by the discussion of the preceding paragraph, the set {$x: V_{\theta}(x) \leq L_{f}L_{v}\epsilon$} is positive invariant and contained in $B(0,\eta)$, and $||x_k||< \epsilon$ implies $V_{\theta}(x_{k+1}) \leq L_{f}L_{v}\epsilon$. Hence $||x_k|| \leq \eta$ for all $k > K$. This completes the proof.

We observe that, within the network structures considered in our experiments, $\eta$ can be made arbitrarily small by reducing $\epsilon$, i.e., we can prove that the state converges to and remains within an arbitrarily small ball centered at the origin.

---

### Decision · Program_Chairs · 2023-09-21

**Decision:**

Accept (poster)

**Comment:**

This paper proposes an approach to learning Lyapunov stable neural network-based controllers for discrete-time systems. The reviewers generally appreciated the novelty of the approach and the strength of the simulation results (though with some desire that they be run at larger scale). While several raised questions about the magnitude of difference between this paper's contribution in the discrete setting vs. prior papers' contributions in the continuous setting, the authors successfully argued that the distinction in the definition of Lyapunov functions between continuous and discrete-time settings yields consequential differences in the actual methodology for learning stable neural network-based controllers, and that their experimental results validate the improved performance of their method against continuous baselines. One reviewer objected to the inclusion of the condition $\|\|x\|\|_\infty \geq \epsilon$ within Lemma 3.1 on theoretical grounds, whereas the authors argued for it on numerical grounds - while this disagreement seems to stem from a difference in perspective between communities, the paper would benefit from further clarification of this as well as discussion on how $\epsilon$ is chosen in practice.